# Toward Undetectable AI Text: AIGT detection evasion with representation editing

## Abstract

With the growing popularity of large language models (LLMs), some concerns have been raised, such as misinformation, plagiarism, and deceptive reviews. Building an efficient and robust AI-generated text (AIGT) detection system has become an urgent demand. To comprehensively assess the robustness of detectors prior to deployment, evasion methods gradually attract the attention of the research community. Existing evasion methods mainly fine-tuned LLMs to align their outputs with human-written text (HWT), which required substantial data and computational resources. Moreover, although leveraging model editing to directly modify the weights of LLMs can significantly reduce the training costs, the evasion performance is not significantly enhanced due to intrinsic limitation of the model-editing theory. To address these limitations, we propose Representation Editing Attack (R-EAT), a training-free evasion method. R-EAT first constructs a difference space between AIGT and HWT. Then, it dynamically edits LLM hidden representation during generation by removing their projections onto this space, thereby encouraging the model to produce more human-like texts. Through theoretical analysis, we demonstrate that R-EAT achieves superior performance by directly editing hidden states, thereby eliminating the inherent limitations of model editing while preserving its advantages in sample and time efficiency. Experimental results demonstrate that the R-EAT effectively reduces the average detection accuracy of 8 AIGT detectors across texts generated by two different LLMs.

## 1 Introduction

Large language models (LLMs) such as deepseek (DeepSeek-AI, 2024), GPT-4 (OpenAI, 2024) and Qwen (QwenTeam, 2025) have made a profound impact on both industrial and academic fields. Despite their impressive performance, LLMs have also raised significant concerns regarding their potential misuse, such as fake news (Su et al., 2024; Hu et al., 2024), academic dishonesty (Wu et al., 2023; Zeng et al., 2024) and deceptive comments designed to manipulate public perception (Mireshghallah et al., 2024). In response to these emerging threats, there is an increasing emphasis on developing robust and reliable methods for detecting AI-generated texts (AIGT) (Mitchell et al., 2023b; Yang et al., 2024; Verma et al., 2024).

Existing AIGT detection methods can be broadly categorized as statistical-based methods (Hans et al., 2024b; Bao et al., 2025; Xu et al., 2025) and classifier-based methods (Tian et al., 2024; Huang et al., 2024; Guo et al., 2024). Although these detectors have achieved strong detection performance, their reliability and robustness in the face of malicious attacks remain to be explored. Therefore, research on AIGT detection evasion methods has drawn widespread interest.

The evasion detection methods aim to decrease the detection probability of AIGT. Previous studies (Krishna et al., 2023; Zhou et al., 2024; Wang et al., 2024; Sadasivan et al., 2025) focus on post-processing methods which required reprocessing each text after generation. While these methods can effectively bypass AIGT detectors, especially when the target detector is accessible, they often degraded text quality and introduced additional computational overhead. Some researchers (Nicks et al., 2023; Pedrotti et al., 2025) have aligned LLM outputs with HWT directly by constructing preference datasets and fine-tuned the models using reinforcement learning techniques such as direct preference optimization (DPO) (Rafailov et al., 2023b). (Pedrotti et al., 2025) first directly fine-tune the LLM with DPO (DPO-1), and then perform another round of DPO fine-tuning based on the

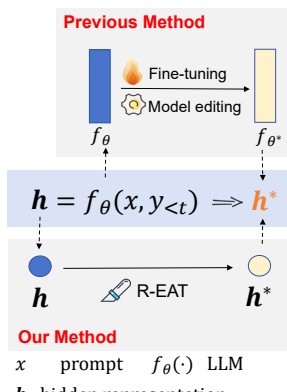

Figure 1: Paradigm of evasion methods.

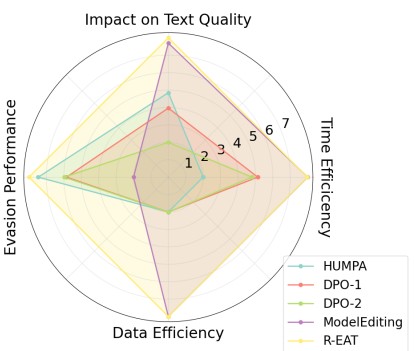

Figure 2: Comparison of evasion methods.

outputs of the DPO-1 model (DPO-2) to further improve performance. As the number of parameters in LLMs continues to rise, the resources needed for fine-tuning them have grown significantly as well. To minimize resource consumption, (Wang et al., 2025) proposed humanized proxy attack (HUMPA), which overrode the predictions of a large model (such as Llama2-13B and Llama3-70B (AI@Meta, 2024)) by using a fine-tuned, smaller humanized model (such as Llama2-7B and Llama3-8B) during decoding. This method achieved remarkable evasion performance avoiding the need to directly fine-tune LLMs with ultra-large-scale parameters. We observe that fine-tuning an 8B model with Low-Rank Adaptation (LoRA) (Hu et al., 2022b) on an NVIDIA A800 GPU still takes about two hours. This significant cost motivates the exploration of more efficient, training-free alternatives. One plausible training-free direction, which we explore for the first time in this work, is to adapt the concept of model editing. As recent studies (Uppaal et al., 2025) in other domains have shown, model editing can alter model behavior by directly modifying the weights of feed-forward networks based on a constructed 'difference space'. However, before even implementing this, our theoretical analysis predicts a critical performance limitation: because of the architecture's residual connections, the output from preceding, unmodified layers is added back to the output of an edited layer. This process effectively dampens or counteracts the intended edits, placing a ceiling on the overall evasion performance. This finding highlights the urgent need for more practical and efficient evasion strategies for AIGT detection evasion.

As shown in Figure 1, we observe that both fine-tuning and model editing ultimately aim to indirectly alter the hidden representations ($h^*$) by modifying the model parameters ($f_\theta$). This observation motivates a more direct and elegant approach: *Instead of navigating the complex and constrained process of parameter modification, why not directly edit the hidden representations $h$ to obtain the desired $h^*$?* Based on this, we propose **R**epresentation **E**diting **AT**tack (**R-EAT**), a training-free detection evasion method based on representation editing. Specifically, we first collect representations of HWTs and AIGTs at layer $j$ of the LLM to construct a difference space between them. During the text generation process, we modify the LLM's hidden representations at layer $j$ by removing their projection onto the difference space, thereby guiding the model to produce more human-like text. As shown in Figure 2, R-EAT significantly improves evasion performance compared to state-of-the-art methods. We conduct theoretical analysis of R-EAT in comparison with existing methods. By leveraging singular value decomposition (SVD) to extract the primary discriminative directions of the difference space, R-EAT not only improves performance but also significantly reduces the data and time consumption required by DPO-based fine-tuning methods. In addition, we model the editing performance of R-EAT and model-editing-based methods to provide a more intuitive comparison between them. R-EAT operates directly on hidden states, thereby eliminating the inherent limitations of model editing. Typically, R-EAT needs to adjust only the last few layers to achieve strong evasion performance, which further reduces total computational costs compared to model editing. Experimental results on Llama2-13b show that R-EAT effectively reduces the average detection accuracy of eight detectors, with decreases of 24.45% and 17.96% on the two datasets, respectively. Our main contributions can be summarized as follows:

- We propose R-EAT, a representation edit-based training-free attack method. It directly modifies hidden representation of LLMs to evade AI-generated text detection, offering a more efficient and cost-effective optimization path compared to previous methods.

- Through theoretical analysis of existing detection evasion methods, we demonstrate that R-EAT achieves stronger performance even with single-layer editing, while requiring neither training nor large amounts of data.

- Experiments on Llama2-13b and Qwen3-14b demonstrate that R-EAT significantly reduces the average detection accuracy across eight detectors on two datasets, while also exhibiting strong advantages in time efficiency, sample requirements, and text quality preservation.

## 2 RELATED WORK

### 2.1 AIGT DETECTION METHODS

Statistical-based methods distinguish HWT from AIGT by exploiting the statistical feature differences between them. DetectGPT (Mitchell et al., 2023a) measured the log probabilities difference between original and perturbed texts. Binoculars (Hans et al., 2024b) calculated the log perplexity of the text using an "observer" LLM, while a "performer" LLM generated next-token predictions, with perplexity determined by the observer's evaluation. Lastde and Lastde++ (Xu et al., 2025) analyzed the differences between the local volatility and global likelihood of the token probability sequence (TPS) in the text. These methods achieved excellent detection performance and demonstrate strong generalization across text generated by various LLMs.

The classifier-based of methods involves training models on large labeled datasets to detect AIGT. Previous work (Guo et al., 2023; Tian et al., 2024) fine-tuned pre-trained language models using different methods. These methods achieved strong detection performance in detecting datasets belonging to the same domain as the training set, but usually failed when faced with datasets that are not in the domain of the training set. To enhance the generalization ability of model in known target domains, Ghostbuster (Verma et al., 2024) took a series of weaker language models as input, performed a structured search over possible feature combinations, and then trained a classifier based on the selected features to improve its generalization. Radar (Hu et al., 2023) employed adversarial learning to jointly train a text paraphraser and an AIGT detector. The paraphraser generates realistic content to evade detection, while the detector leverages feedback from the paraphraser to optimize itself, significantly enhancing its robustness against LLM paraphrasing attacks. DP-Net (Zhou et al., 2025) enhanced the generalization and robustness of the detector by adding dynamic perturbations to the text embeddings during training to simulate domain shift scenarios. (Chakraborty et al., 2023) have thoroughly analyzed the possibilities of AIGT detection and theoretically demonstrated that, as long as the distributions of AIGTs and HWTs are not exactly the same, detection remains feasible by simply increasing the number of samples.

### 2.2 DETECTION EVASION METHODS

To reveal vulnerabilities in AI detectors before they are deployed in real-world applications, (Krishna et al., 2023) fine-tuned T5 (Raffel et al., 2020) enabling the model to perform diverse modifications and paraphrasing of text without altering its original meaning. (Zhou et al., 2024) performed synonym replacement on the important token in text with highest score until it can not be detected by the proxy detector. These methods require individual modifications for each text, resulting in substantial time consumption. To encourage LLMs to generate human-like texts directly, some researches (Nicks et al., 2023; Pedrotti et al., 2025) fine-tuned the LLM using direct preference optimization. Furthermore, (Wang et al., 2025) proposed Humanized proxy attack (HUMPA), which fine-tunes a proxy small model using DPO and modifies the output probabilities of the target LLM based on the probability changes before and after fine-tuning the proxy model. This method achieved remarkable evasion performance while significantly reducing significantly resource consumption required for fine-tuning LLMs with large-scale parameters. However, this method essentially does not address the resource consumption caused by fine-tuning. Therefore, we propose R-EAT, a training-free detection evasion method. Its detail will be introduced in the following sections.

## 3 METHOD

### 3.1 TASK DEFINITION

**AIGT Detection.** The task aims to classify whether a given text sequence $y = [y_1, \ldots, y_T] \in \mathcal{Y}$ is AI-generated or human-written. Given a detector $M$, the probability that it assigns to the text sequence $y$ being generated by AI is $P_M(y)$. If $P_M(y)$ is greater than a threshold $\gamma$, the detector classifies the text as AI-generated; otherwise, it classifies it as human-written.

**Detection Evasion.** We denote the generative processes of AIGT and HWT as $\pi_{AI}$ and $\pi_H$, respectively. Given a prompt $x \in \mathcal{X}$, the goal of the task is to find a new generation strategy $\pi_{AI}^*$, such that $\pi_{AI}^*(y|x)$ is as close as possible to $\pi_H(y|x)$. Formally, the task is to solve the optimization problem:

$$\pi^* = \arg \min_{\pi_{AI}} \sum_{y \in \mathcal{Y}} \pi_{AI}(y|x) \log \frac{\pi_{AI}^*(y|x)}{\pi_H(y|x)}. \tag{1}$$

### 3.2 R-EAT

To compensate for the differences between AIGT and HWT, we propose R-EAT, a training-free detectionn evasion method. R-EAT first constructs a difference space between AIGT and HWT at layer $j \in [1, 2, \ldots, L]$, where $L$ is the total number of layers in the target LLM. During text generation, we directly modify the hidden state of the layer $j$ to steer the LLM toward producing more human-like text.

**Difference Space Construction.** To construct the difference space between HWT and AIGT, we first build a preference dataset $D := \{(x_i, y_i^+, y_i^-)\}_{i=1}^N$, where $y_i^+ \sim \pi_H(\cdot|x_i)$, $y_i^- \sim \pi_{AI}(\cdot|x_i)$ and $N$ is the number of samples. For each $y_i^+ \in \mathcal{Y}^+$ and $y_i^- \in \mathcal{Y}^-$, we feed them into the target LLM to extract the hidden representations of last token from layer $j$, denoted as $\boldsymbol{h}_{i,j}^+ \in \mathbb{R}^d$ and $\boldsymbol{h}_{i,j}^- \in \mathbb{R}^d$, respectively. We stack these representations respectively as $\boldsymbol{H}_j^+, \boldsymbol{H}_j^- \in \mathbb{R}^{N \times d}$, and the difference matrix $\boldsymbol{Z}_j \in \mathbb{R}^{N \times d}$ is calculated as:

$$\boldsymbol{Z}_j = \boldsymbol{H}_j^- - \boldsymbol{H}_j^+. \tag{2}$$

As shown in Appendix B, we find that the mean directions of $\boldsymbol{H}_j^+$ and $\boldsymbol{H}_j^-$ are aligned with their respective un-centered top right singular vectors, and that these mean directions are also aligned with each other. Therefore, we assume that the $\boldsymbol{H}_j^+$ and $\boldsymbol{H}_j^-$ share the same mean direction. To focus on the directions that are orthogonal to the overall human representation, we remove the component along the mean direction of $\boldsymbol{H}_j^+$. Specifically, given the mean vector $\bar{\boldsymbol{h}}_j = \frac{1}{N} \sum_{i=1}^N \boldsymbol{h}_{i,j}^+$, the centered difference matrix $\tilde{\boldsymbol{Z}} \in \mathbb{R}^{N \times d}$ is calculated as:

$$\tilde{\boldsymbol{Z}}_j = \boldsymbol{Z}_j (\boldsymbol{I} - \frac{\bar{\boldsymbol{h}}_j (\bar{\boldsymbol{h}}_j)^\top}{||\bar{\boldsymbol{h}}_j||_2^2}), \tag{3}$$

Finally, we apply singular value decomposition (SVD) on to extract the most discriminative components between HWT and AIGT:

$$\tilde{\boldsymbol{Z}}_j = \boldsymbol{U} \boldsymbol{S} \boldsymbol{V}^\top, \tag{4}$$

where $\boldsymbol{U} \in \mathbb{R}^{N \times d}$ is the left singular matrix, $\boldsymbol{S} \in \mathbb{R}^{d \times d}$ is diagonal matrix containing the singular values and $\boldsymbol{V} \in \mathbb{R}^{d \times d}$ is the right singular matrix.

We select the all right singular vectors whose cumulative explained variance exceeds a threshold $\tau = 90\%$, and use them to form the difference space basis matrix at layer $j$:

$$\boldsymbol{B}_j = [\boldsymbol{v}_1, \boldsymbol{v}_2, \ldots, \boldsymbol{v}_i, \ldots, \boldsymbol{v}_k] \in \mathbb{R}^{d \times k}, \tag{5}$$

where $\boldsymbol{v}_i$ represents the $i_{th}$ right singular vector and $d$ is the dimension of the hidden state.

**Representation Editing.** After obtaining the difference space for the layer $j$, we modify the hidden state of the last token at the layer $j$ during text generation. Specifically, during the text generation

Figure 3: Case study of the text generated after employing the R-EAT and model editing on the Llama2-13b. The yellow highlighted parts indicate garbled text.

process of the LLM, for a given prompt $x_{new}$, we first extract the hidden state of the last token $\boldsymbol{h}_{new,j}$ at the layer $j$. And then, we remove its projection onto the difference space $\boldsymbol{B}_j$, leading to a modified hidden representation:

$$\hat{\boldsymbol{h}}_{new,j} = (\boldsymbol{I} - \alpha \boldsymbol{B}_j(\boldsymbol{B}_j)^\top)\boldsymbol{h}_{new,j}, \qquad (6)$$

where $\alpha$ denotes the hyperparameter controlling the editing strength. The language model then continues generating text based on $\hat{\boldsymbol{h}}_{new,j}$.

### 3.3 ANALYSIS

In this section, we provide a theoretical comparison between R-EAT and existing fine-tuning-based and model-editing-based methods, and reveal the advantages of R-EAT. Previous detection evasion methods used DPO to fine-tune LLM to align their outputs with HWTs. As demonstrated in (Uppaal et al., 2025), the DPO objective inherently involves modeling the discrepancy between two hidden states. In contrast to DPO, SVD directly computes the principal components of the difference space via a low-rank decomposition, achieving an optimal low-rank approximation with greater sample efficiency and faster convergence. Based on this insight, (Uppaal et al., 2025) indicated that once the difference space is established, model editing offers a training-free alternative to attain effects comparable to DPO. We model the editing performance of detection evasion methods to provide a more intuitive comparison between model editing and R-EAT. Given a selected set of layers $\mathcal{S}$ to be edited, the total editing performance $E_m(\mathcal{S})$ and $E_r(\mathcal{S})$ for model editing and R-EAT are given by:

$$E_m(\mathcal{S}) \approx \sum_{j \in \mathcal{S}} e_j^m, \quad E_r(\mathcal{S}) \approx \sum_{j \in \mathcal{S}} e_j^r, \qquad (7)$$

where $e_j^m$ and $e_j^r$ represent the editing performance of employing model editing and R-EAT at layer $j$ on the model's final detection-evasion performance, respectively. When the modifications introduced at each edited layer are sufficiently small, their effects on the final hidden representation pass through the residual connections and can be well approximated by first-order Taylor expansion. As a result, the overall editing performance $E(\mathcal{S})$ can be regarded as the approximate linear accumulation of the per-layer contributions. The detailed derivation is provided in Appendix C.1.

Consider a single layer in transformer-based LLMs, the hidden representation $\boldsymbol{h}_j$ at layer $j$ is obtained by:

$$\boldsymbol{h}_j = \boldsymbol{h}'_{j-1} + FFN(LN(\boldsymbol{h}'_{j-1})), \qquad (8)$$

$$\boldsymbol{h}'_{j-1} = \boldsymbol{h}_{j-1} + MHA(LN(\boldsymbol{h}_{j-1})), \qquad (9)$$

where $\boldsymbol{h}_{j-1}$ represents the hidden representation from the previous layer. $\boldsymbol{h}'_{j-1}$ denotes the output of applying multi-head attention ($MHA(\cdot)$) with a residual connection to $\boldsymbol{h}_{j-1}$. $LN(\cdot)$ denotes the layer normalization function. $FFN(\cdot)$ is a feed-forward network, defined as:

$$FFN(\boldsymbol{h}'_{j-1}) = \boldsymbol{W}_{2,j}\boldsymbol{z}_j = \boldsymbol{W}_{2,j}(\sigma(\boldsymbol{W}_1 LN(\boldsymbol{h}'_{j-1}))), \qquad (10)$$

where $\sigma$ is an activation function, $\boldsymbol{W}_1$ and $\boldsymbol{W}_{2,j}$ are weight matrices. According to key-value theory (Geva et al., 2020), the model edit-based methods is aligned with the target through by directly editing $\boldsymbol{W}_{2,j}$. In contrast, our R-EAT directly edits the hidden representation $\boldsymbol{h}_j$. The hidden representations $\hat{\boldsymbol{h}}_j^m$ and $\hat{\boldsymbol{h}}_j^r$, resulting from applying model editing and R-EAT respectively to the layer $j$, are formally defined as:

$$\hat{\boldsymbol{h}}_j^m = \boldsymbol{h}_{j-1}' + (\boldsymbol{I} - \boldsymbol{P}_j)\boldsymbol{W}_{2,j}\boldsymbol{z}_j, \quad \hat{\boldsymbol{h}}_j^r = (\boldsymbol{I} - \boldsymbol{P}_j)(\boldsymbol{h}_{j-1}' + \boldsymbol{W}_{2,j}\boldsymbol{z}_j), \tag{11}$$

where $\boldsymbol{P}_j = \boldsymbol{B}_j(\boldsymbol{B}_j)^\top$ refers to the projection matrix of the difference space at layer $j$. Eq. 11 reveals a fundamental flaw in model-editing: edits applied to the feed-forward network are consistently diluted by the unmodified signal reintroduced via the residual connection. This architectural bottleneck places a hard ceiling on the method's overall evasion capabilities. We employ the ratio $R$ of $\boldsymbol{e}_j^r$ and $\boldsymbol{e}_j^m$ to further compare the editing performance of these two methods on single layer, which is defined as:

$$R = \frac{||\boldsymbol{e}_j^r||_F}{||\boldsymbol{e}_j^m||_F} \approx \frac{||\boldsymbol{h}_j - \hat{\boldsymbol{h}}_j^r||_F}{||\boldsymbol{h}_j - \hat{\boldsymbol{h}}_j^m||_F} = \frac{||\boldsymbol{P}_j(\boldsymbol{h}_{j-1}' + \boldsymbol{W}_{2,j}\boldsymbol{z}_j)||_F}{||\boldsymbol{P}_j\boldsymbol{W}_{2,j}\boldsymbol{z}_j||_F} \propto \sqrt{j}. \tag{12}$$

where $\|\cdot\|_F$ is Frobenius norm (Böttcher & Wenzel, 2008). The detailed derivation is provided in Appendix C.2. From Eq. 12, it can be observed that, within the same layer, editing the hidden state has a more significant effect than editing the weight matrix $\boldsymbol{W}_{2,j}$. Additionally, R-EAT becomes increasingly more effective in deeper layers. Moreover, it typically requires modifying only the last few layers to achieve a similar effect, which leads to lower overall computational cost and a smaller impact on text quality compared to model editing, as illustrated in Figure 3. We provide a comprehensive experimental study in the next section to evaluate the effectiveness of R-EAT.

## 4 EXPERIMENTS

### 4.1 DATASETS

We conduct main experiments on two datasets, including OpenWebText (Gokaslan et al., 2019) and WritingPrompts (Fan et al., 2018). To verify the effectiveness of our method, we also conduct additionally experiments on PubMedQA Jin et al. (2019) and SQuAD (Rajpurkar et al., 2016). The detailed results can be found in the Appendix F.1. We randomly select 3,000 human-written texts from each dataset. Their first 8 tokens served as prompts for the LLM to generate the corresponding AI-generated texts. The data were split into training, validation, and test sets in a 7:1.5:1.5 ratio. For fine-tuning-based methods, the entire training set was used. In contrast, for R-EAT, only 500 sentences from the training set were selected to construct the difference space. We evaluate the impact of sample size on R-EAT performance, and the corresponding results are presented in section 4.7.

### 4.2 DETECTORS

We conduct experiments using eight detectors, which can be grouped into two main categories: **(1) Classifier-based detector**, including RoBERTa-base and RoBERTa-large (Solaiman et al., 2019). **(2) Statistical-based detector**, including Likelihood (Solaiman et al., 2019), Entropy (Gehrmann et al., 2019), DetectLRR (Su et al., 2023), Binoculars (Hans et al., 2024b), Lastde and Lastde++ (Xu et al., 2025). Details of these detectors can be found in the Appendix D.

### 4.3 BASELINES

We employ four evasion methods as baselines to evaluate the effectiveness of R-EAT, including **(1) DPO-1** (Pedrotti et al., 2025) directly fine-tunes the target LLM using DPO. **(2) DPO-2** (Pedrotti et al., 2025) uses the outputs generated by the DPO-1 fine-tuned LLM to perform a second DPO fine-tuning, further refining the model. **(3) HUMPA** (Wang et al., 2025) first fine-tunes a small language model using DPO as a proxy. During text generation, the proxy model is used to adjust the logits of the target LLM, guiding it to produce more human-like text. **(4) ModelEditing** (Uppaal et al., 2025) treats model editing as robust denoising with a single-step DPO. We introduce it to the evasion detection task for the first time and use it as a baseline for comparison. Implementation details of these Methods can be found in the Appendix E.

Table 1: Comparison of AUROC across different detectors on OpenWebText and WritingPrompt using Llama2-13b. The best results are highlighted in bold.

| Datasets | Detector | Base | HUMPA | DPO-1 | DPO-2 | ModelEditing | R-EAT |
|---|---|---|---|---|---|---|---|
| OpenWebText | RoBERTa-base | 0.9808 | 0.9616 | 0.8569 | 0.8362 | 0.9726 | **0.8004** |
| | RoBERTa-large | 0.9670 | 0.9609 | 0.8899 | 0.8811 | 0.9670 | **0.7831** |
| | Likelihood | 0.9537 | **0.0667** | 0.5266 | 0.5355 | 0.8692 | 0.6613 |
| | Entropy | 0.3527 | 0.9520 | 0.6355 | 0.6052 | 0.4927 | **0.2534** |
| | DetectLRR | 0.9688 | **0.2406** | 0.6094 | 0.6237 | 0.9458 | 0.7217 |
| | Binoculars | 0.8998 | 0.8789 | 0.9233 | 0.8934 | 0.9287 | **0.7504** |
| | Lastde | 0.9782 | 0.6122 | 0.7099 | 0.7432 | 0.9622 | **0.7589** |
| | Lastde++ | 0.9793 | 0.6178 | 0.6459 | 0.6366 | 0.8663 | **0.3949** |
| | Average | 0.8850 | 0.6613 | 0.7247 | 0.7194 | 0.8756 | **0.6405** |
| WritingPrompt | RoBERTa-base | 0.9715 | 0.9662 | 0.8558 | 0.8680 | 0.9873 | **0.6991** |
| | RoBERTa-large | 0.9437 | 0.9246 | 0.8918 | 0.8757 | 0.9524 | **0.6604** |
| | Likelihood | 0.9843 | 0.8468 | **0.5536** | 0.5578 | 0.9694 | 0.8326 |
| | Entropy | 0.1260 | 0.5336 | 0.5534 | 0.5317 | 0.3615 | **0.1613** |
| | DetectLRR | 0.9864 | 0.9096 | **0.5703** | 0.5914 | 0.9775 | 0.8314 |
| | Binoculars | 0.9389 | 0.9017 | 0.8673 | 0.8610 | 0.9102 | **0.7376** |
| | Lastde | 0.9990 | 0.9921 | **0.7042** | 0.7188 | 0.9928 | 0.8977 |
| | Lastde++ | 0.9905 | 0.9637 | 0.6399 | **0.6110** | 0.9846 | 0.6827 |
| | Average | 0.8675 | 0.8798 | 0.7045 | 0.7019 | 0.8920 | **0.6879** |

## 4.4 EVALUATION METRICS

Following previous work, we evaluate the effectiveness of the methods using the area under the receiver operating characteristic curve (AUROC). We also report the area under the precision-recall curve (AUPRC), with the results provided in the appendix F.5. To evaluate the impact of detection evasion methods on text generation quality, we report BERTScore-F1 (Zhang et al., 2019), $|\Delta\text{perplexity}|$ ($|\Delta\text{PPL}|$), and $|\Delta\text{Entropy}|$, where $\Delta$ denotes the difference in each metric between texts generated before and after applying the methods. Additionally, we invite 5 participants with extensive experience in using LLMs to rate the generated texts in terms of fluency, semantic accuracy, and the probability of being AI-generated.

## 4.5 IMPLEMENTATION SETTING

We conduct experiments using Llama-2-13b-chat-hf (Touvron et al., 2023) and Qwen3-14b (Qwen-Team, 2025). In HUMPA, we use Llama-2-7b-chat-hf and Qwen3-8b as the proxy small models, respectively. The experimental results of qwen3-8b are provided in Appendix F.4. Additionally, we construct the difference space using the right singular vectors whose cumulative variance contribution exceeds $\tau = 90\%$. In our main experiments, edits are applied only to the last three layers of the target LLM, and the editing strength $\alpha = 0.7$. All experiments are conducted on a single NVIDIA A800 GPU.

## 4.6 EXPERIMENTAL RESULTS

**Detection Evasion Performance.** Table 1 shows the AUROC scores across eight detectors for Llama2-13b on the OpenWebText dataset and WritingPrompt dataset after applying different detection-evasion methods. It can be observed that R-EAT achieves the highest average evasion performance on both datasets compared to other methods. Notably, applying R-EAT reduces the average detection accuracy of generated text from 88.50% to 64.05% on the OpenWebText dataset. Additionally, we observe that R-EAT is less effective than other methods on some statistical-based detectors including Likelihood and DetectLRR across both datasets. We argue that this is because fine-tuning updates all parameters across layers, thereby inducing a global shift in token selection probabilities. Meanwhile, we find that R-EAT consistently outperforms other methods

Table 2: Llama-generated text quality evaluation on OpenWebText dataset. In the table, $flu.$, $sem.$, and $prob.$ are human evaluation metrics, representing fluency, semantic accuracy, and the probability of being AI-generated, respectively. The best results are highlighted in bold, while the second-best results are underlined.

| Methods | BERTscore↑ | \|ΔPPL\|↓ | \|ΔEntropy\|↓ | $flu.$ ↑ | $Sem.$ ↑ | $prob.$ ↓ |
|---------|-----------|-----------|--------------|----------|----------|-----------|
| DPO-1 | 0.8068 | 93.0307 | 1.2379 | 3.5438 | **3.4638** | 2.8125 |
| DPO-2 | 0.8065 | 118.6315 | 0.4756 | **3.6000** | 3.4438 | 2.7612 |
| HUMPA | 0.7819 | 93.0307 | 1.2379 | 2.8075 | 2.8225 | 3.1887 |
| ModelEdit | **0.7884** | 10.5727 | **0.0938** | 3.4025 | 3.3013 | 2.6238 |
| R-EAT | 0.7595 | **1.7057** | 0.4206 | 3.5336 | 3.4513 | **2.5188** |

against classifier-based detectors, indicating that R-EAT indeed alters the global distributional properties of the generated text. Figure 4 illustrates that applying R-EAT shifts the distribution of Llama-generated texts toward greater alignment with HWTs. Further experiments on additional detectors, including Fast-detectGPT (Bao et al., 2023), Radar (Hu et al., 2023), Text Fluoroscopy Yu et al. (2024)and ImBD Chen et al. (2025), are presented in Appendix F.2. To evaluate the generalization of the difference space across different corpora, we construct the difference space on Llama2/OpenWebText and apply it to guide text generation on the WritingPrompts dataset (refer to R-EAT$_{ood}$). The results show that R-EAT$_{ood}$ even outperforms the baseline method trained directly on WritingPrompts, with performance only slightly lower than constructing the space using WritingPrompts itself. These results show that R-EAT has strong generalization ability. Detailed experimental results can be found in the Appendix F.3.

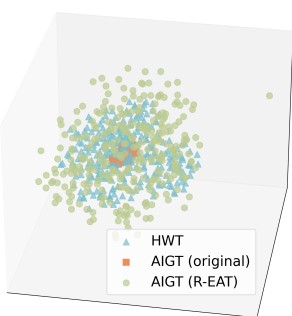

Figure 4: T-SNE visualization.

**Evaluation of Generated Text Quality.** We evaluate text quality using both automatic metrics and human judgments. In the human evaluation experiment, we collecte 100 texts modified by each evasion method and asked 8 experienced LLM users to assess them along three dimensions: fluency, semantic clarity, and likelihood of AI generation. The ratings ranged from 1 to 5, where a higher fluency score indicates smoother text, a higher semantic score reflects better interpretability, and a higher AI-probability score denotes a greater chance of being recognized as AI-generated. As shown in Table 2, R-EAT outperforms the three fine-tuned baselines on all automatic metrics, indicating its effectiveness in mitigating the negative impact on the quality of AIGT. Additionally, we find that although model editing achieves the best scores on BERTScore and |ΔEntropy|, its performance is rated worse in human evaluation compared to R-EAT.

**Time Efficiency.** We evaluate the time cost of different methods. For the three fine-tuned approaches, we measure the training time, while we report the time required to construct the difference space for ModelEditing and R-EAT. As shown in Figure 5, R-EAT substantially reduces time consumption, significantly improving the efficiency compared to existing detection evasion methods. In addition, Figure 5 also indicates that HUMPA, despite fine-tuning only a small proxy model, can incur higher time costs than directly fine-tuning the LLM (DPO-1, DPO-2). This is because we set the overall number of training epochs for HUMPA to be three times that of DPO in our experiments to achieve better detection evasion. Detailed implementation settings for these baselines can be found in Appendix E.

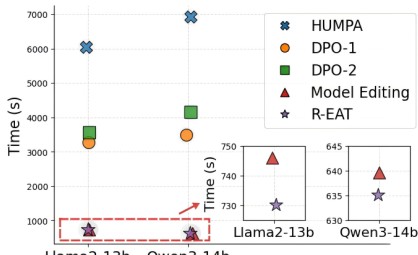

Figure 5: Time consumption experiments on the OpenWebText dataset.

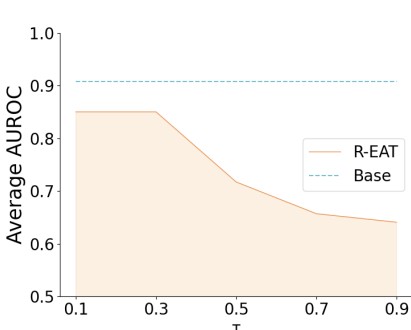

Figure 6: Impact of $\tau$.

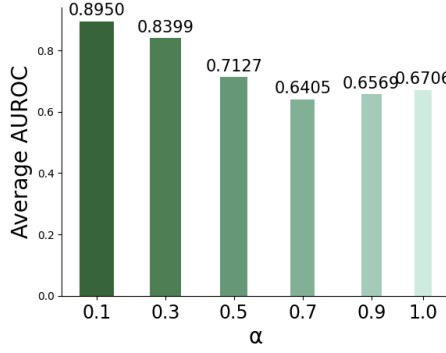

Figure 7: Impact of $\alpha$.

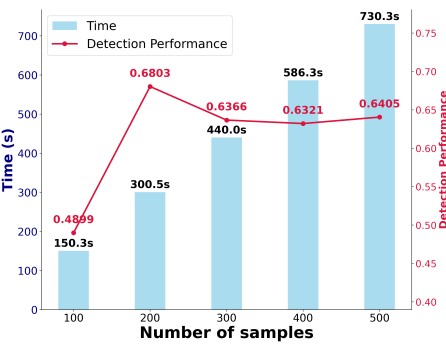

Figure 8: Comparison of cost time and evasion performance across different sample sizes.

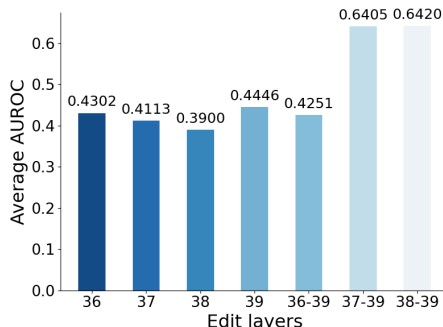

Figure 9: Impact of number of layers edited.

### 4.7 HYPERPARAMETER ANALYSIS

We conduct hyperparameter analysis to study the sensitivity of our method to different settings.

**Impact of the cumulative variance threshold $\tau$.** An increase in $\tau$ leads to a greater number of singular vectors used to construct the difference space. As shown in Figure 6, this allows the space to capture the distinctions between HWT and AIGT more accurately, which in turn improves evasion performance.

**Impact of editing strength $\alpha$.** This parameter $\alpha$ controls the proportion of the target directional component removed from original hidden representation. As depicted in the Figure 7, increasing $\alpha$ in R-EAT initially enhances evasion performance, as larger hidden-state modifications push generated text further from AI features and closer to human-like style. However, beyond a certain value (e.g., when $\alpha = 0.7$), increasing $\alpha$ degrades the semantic quality of the generated text, leading to a decline in performance. We evaluate the PPL of the generated texts at different editing strengths, which supports our hypothesis, as shown in the Appendix F.5.

**Impact of the number of samples.** As shown in Figure 8, increasing the sample size gradually increases the time needed to build the space. Additionally, we find that constructing the difference space with only 100 samples is sufficient to reduce the average detection accuracy of eight detectors to 48.99%. However, as the sample size increases, the evasion performance starts to degrade. As provided in Appendix F.7, we find that the number of singular vectors satisfying $\tau > 90\%$ increases with sample size, indicating a rise in the complexity of the difference space. Additionally, the cosine of the principal angle between the difference space constructed with 200 samples and the one built with 100 samples is the lowest, suggesting that larger sample sizes introduce more noise and suboptimal features. This results in a decline in evasion performance. Although the performance

begins to recover with larger sample sizes, it still does not reach the optimal level observed with 100 samples.

**Impact of the number of layers edited.** As illustrated in Figure 9, when editing a single layer, the performance of the edit improves as the layer index increases. For instance, editing layer 36 reduces the average detection rate to 43.02%, while editing layer 38 further lowers it to 39%. This is because when the edited hidden vectors are passed to the subsequent layers, they are still influenced by the model features encoded in the deeper layers, which ultimately degrades the evasion performance. The effect is more pronounced when the edited layer is closer to the beginning of the model. Interestingly, when only the last layer's hidden vectors are edited, the performance actually worsens. We believe this occurs because the last layer mainly maps hidden representations to output probability distributions, and modifying it alone only slightly adjusts the output without significantly altering the model's generation strategy, which limits the effectiveness of the evasion. Additionally, for multi-layer editing, a clear trend emerges: the more layers that are edited, the better the evasion performance, which aligns with the cumulative effect observed in Section 3.3. However, we also observe that editing both layers 38 and 39 results in worse performance compared to editing only a single layer. We attribute this phenomenon to the large edits introduced at layer 38, which cause higher-order terms in the nonlinear transformations to become significant. This aligns with the remark made in Appendix C.1. Experiments with a small editing strength ($\alpha = 0.5$) are also performed to validate our analysis, with results provided in the Appendix F.8.

## 5 CONCLUSION

In this paper, we propose R-EAT, which is a novel, training-free method for evading detection through representation editing. By constructing a difference space between the hidden states of AIGT and HWT at the layer $j$ and removing their projections onto this space, R-EAT effectively encourages the LLM to produce more human-like text. Compared with prior methods, R-EAT delivers strong detection evasion performance with remarkable time and sample efficiency, achieving substantial improvements using only 500 samples. Additionally, we suggest several potential strategies to defend against R-EAT, which are presented in the appendix G.

## ETHICS STATEMENT

This work investigates methods for evading AI-generated text detection. Our goal is strictly research-focused: to understand model behaviors and identify limitations in current detection techniques, ultimately guiding the development of more robust detectors. We emphasize that our work is not intended to facilitate malicious use of detection evasion techniques. Instead, the primary contribution lies in providing insights into the weaknesses of existing detectors, thereby motivating the design of stronger detection mechanisms. We believe that research on detection evasion can help the community better understand the limitations of current detectors and ultimately contribute to the development of more robust and secure AI systems. Our findings should therefore be viewed as a step toward improving AI safety, fairness, and trustworthiness, rather than as a means of circumventing responsible deployment practices.

## REPRODUCIBILITY STATEMENT

The code required to reproduce all experiments has been submitted as supplementary material. The appendix contains: (i) the LLM usage statement A, (ii) derivations of the formulas presented in the paper C, (iii) the implementation details of the detectors D, (iv) the implementation details of baselines E, (v) all additional experimental results F, and (vi) potential defense strategies G.

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

## A    LLM Usage Statement

In preparing this manuscript, we use chatGPT OpenAI (2024) for language polishing and stylistic refinement. All scientific content, experimental design, analysis, and conclusions were independently developed by the authors.

## B    Explanation around the centering step

Following (Uppaal et al., 2025), we selected 500 samples each from AIGT and HWT, and used Llama2-13b to compute the hidden states of the last token for each sample at each layer. We then calculate the cosine similarity between the average hidden state of each sample class and its first right singular vector, as well as the cosine similarity between the average hidden states of the two classes. As shown in the Figure 10, we find that the majority of the information in both the HWT and AIGT distributions is concentrated along their respective mean directions, with the mean directions of the two distributions nearly coinciding. Based on this, we assume that HWT and AIGT share the same mean direction.

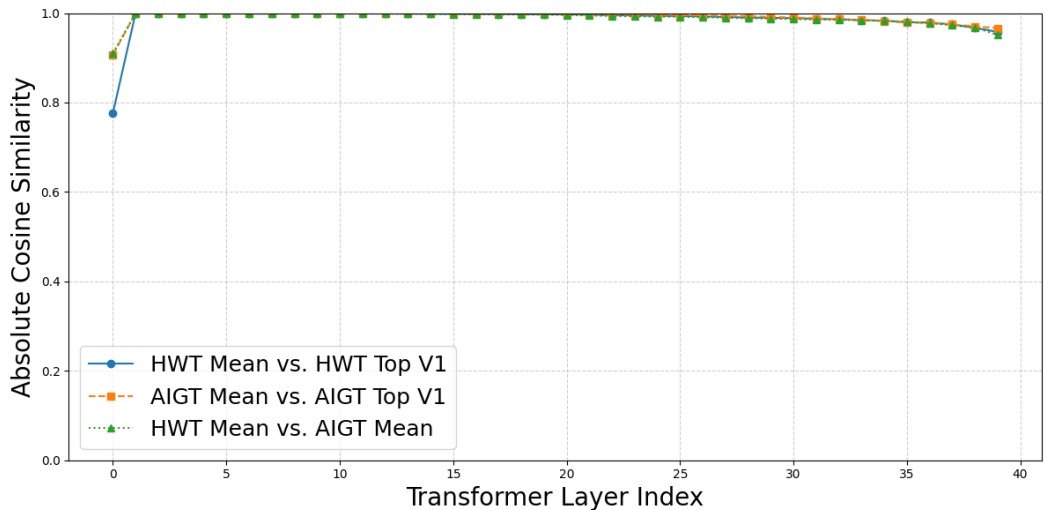

Figure 10: Layer-wise changes in the absolute cosine similarity.

## C  FORMULA DERIVATION

### C.1  LINEAR ACCUMULATION OF EDITING PERFORMANCE WITH QUADRATIC REMAINDER

Let $F_j : \mathbb{R}^d \to \mathbb{R}^d$ $(j = 1, \ldots, L)$ be the layer-wise mappings of a transformer, and the overall mapping of the transformer is then given by the composition $F = F_L \circ F_{L-1} \circ \cdots \circ F_1$. Assume each $F_j$ is twice continuously differentiable $(C^2)$ on a convex neighborhood containing the nominal hidden states $\boldsymbol{h}_j$, and that there exists $M > 0$ such that for every relevant point the operator norm of every second derivative satisfies

$$\|D^2 F_j(\cdot)\|_{\mathrm{op}} \le M \tag{13}$$

where $\| \cdot \|_{\mathrm{op}}$ is the operator norm. $D^2 F_j$ denotes the second derivative of $F_j$.

**(1) R-EAT.** Let $\mathcal{S} \subset \{1, \ldots, L\}$. For each $j \in \mathcal{S}$, introduce a modification $\delta\boldsymbol{h}_j \in \mathbb{R}^d$ to the output of layer $j$. The resulting final representation is $\hat{\boldsymbol{h}}_L^r$, and the total editing performance $E_r S$ can be expressed as

$$E_r(S) := \hat{\boldsymbol{h}}_L^{\,r} - \boldsymbol{h}_L = \sum_{j \in \mathcal{S}} J_{j \to L}\, \delta\boldsymbol{h}_j + R \approx \sum_{j \in \mathcal{S}} \boldsymbol{e}_j^r, \tag{14}$$

where $J_{j \to L} := D_{\boldsymbol{h}_j}\big(F_L \circ \cdots \circ F_{j+1}\big)(\boldsymbol{h}_j)$ is the Jacobian mapping from $\boldsymbol{h}_j$ to $\boldsymbol{h}_L$. $\boldsymbol{e}_j^r$ refers to the edit performance on layer $j$. The remainder $R$ satisfies the quadratic bound:

$$\|R\| \le \frac{C_r}{2} \Big( \sum_{j \in \mathcal{S}} \|\delta\boldsymbol{h}_j\| \Big)^2, \tag{15}$$

where $C_r > 0$ depending only on $L$ and the per-layer second-derivative bound of $F_j$.

**(2) Model editing.** For $j \in \mathcal{S}$, let $\Delta\boldsymbol{W}_j$ be a small modification of the parameters of layer $j$, and let $\mathrm{vec}(\Delta\boldsymbol{W}_j)$ denote its vectorization. The total editing performance $E_m(S)$ is given by:

$$E_m(S) := \hat{\boldsymbol{h}}_L^m - \boldsymbol{h}_L = \sum_{j \in \mathcal{S}} J_{j \to L}^{\boldsymbol{W}}\, \mathrm{vec}(\Delta\boldsymbol{W}_j) + R' \approx \sum_{j \in \mathcal{S}} \boldsymbol{e}_j^m, \tag{16}$$

where $J_{j \to L}^{\boldsymbol{W}} := D_{\boldsymbol{h}_j}(F_L \circ \cdots \circ F_{j+1})(\boldsymbol{h}_j)\, D_{\boldsymbol{W}_j} \boldsymbol{h}_j(\boldsymbol{W}_j)$ maps weight perturbations to the final representation, and the remainder satisfies

$$\|R'\| \le \frac{C_m}{2} \Big( \sum_{j \in \mathcal{S}} \|\mathrm{vec}(\Delta\boldsymbol{W}_j)\| \Big)^2, \tag{17}$$

where $C_m > 0$ depending on $L$ and the second-derivative bounds of both the layer mappings $F_j$ and the hidden-state function $\boldsymbol{h}_j(\boldsymbol{W}_j)$.

**Proof.** For each layer index $j$, we define the downstream composition $G_j := F_L \circ F_{L-1} \circ \cdots \circ F_{j+1}$, so that $\boldsymbol{h}_L = G_j(\boldsymbol{h}_j)$ for any $j$. By assumption each $G_j$ is $C^2$ on the relevant neighborhood, and its second derivative operator norm is bounded by a constant $M_G$, which depends on $M$ and $L$.

*Single layer editing.* Fix a single layer $j \in \mathcal{S}$ and consider a small editing $\delta\boldsymbol{h}_j$ applied to its hidden representation. By applying a second-order Taylor expansion of $G_j$ around $\boldsymbol{h}_j$, the edited output can be expressed as:

$$G_j(\boldsymbol{h}_j + \delta\boldsymbol{h}_j) = G_j(\boldsymbol{h}_j) + D_{\boldsymbol{h}_j} G_j(\boldsymbol{h}_j)\,\delta\boldsymbol{h}_j + \frac{1}{2}\,\delta\boldsymbol{h}_j^\top H_j(\xi_j)\,\delta\boldsymbol{h}_j, \tag{18}$$

where $\xi_j$ lies on the line segment between $\boldsymbol{h}_j$ and $\boldsymbol{h}_j + \delta\boldsymbol{h}_j$, and $H_j(\xi_j)$ denotes the second Fréchet derivative of $G_j$ at $\xi_j$. Taking norms and using $\|H_j(\xi_j)\|_{\mathrm{op}} \leq M_G$ yields:

$$\big\|G_j(\boldsymbol{h}_j + \delta\boldsymbol{h}_j) - G_j(\boldsymbol{h}_j) - D_{\boldsymbol{h}_j} G_j(\boldsymbol{h}_j)\,\delta\boldsymbol{h}_j\big\| \leq \tfrac{M_G}{2}\,\|\delta\boldsymbol{h}_j\|^2. \tag{19}$$

*Multi-layers editing.* Consider editing layers in order from the deepest (largest index) toward the shallowest (smallest index). Define a vector-valued function $\Phi : \prod_{j \in \mathcal{S}} \mathbb{R}^d \to \mathbb{R}^d$ that maps the set of layer-wise perturbations $(\delta\boldsymbol{h}_j)_{j \in \mathcal{S}}$ to the final representation as follow:

$$\Phi\big((\delta\boldsymbol{h}_j)_{j \in \mathcal{S}}\big) := F_L \circ \cdots \circ F_1 \Big|_{\boldsymbol{h}_j \mapsto \boldsymbol{h}_j + \delta\boldsymbol{h}_j,\, j \in \mathcal{S}}. \tag{20}$$

Since $\Phi$ is $C^2$ in the joint variables, a second-order multivariate Taylor expansion about zero perturbation is given by:

$$\Phi(\boldsymbol{0} + \boldsymbol{\delta}) = \Phi(\boldsymbol{0}) + \sum_{j \in \mathcal{S}} D_j \Phi(\boldsymbol{0})[\delta\boldsymbol{h}_j] + \tfrac{1}{2} \sum_{p,q \in \mathcal{S}} \langle H_{pq} \delta\boldsymbol{h}_p, \delta\boldsymbol{h}_q \rangle, \tag{21}$$

where $D_j \Phi(\boldsymbol{0})[\delta\boldsymbol{h}_j] = J_{j \to L} \delta\boldsymbol{h}_j$ is the partial Fréchet derivative. $\{H_{pq}\}_{p,q \in \mathcal{S}}$ are the mixed second derivatives. $\langle \cdot, \cdot \rangle$ denotes the standard Euclidean inner product. By applying operator-norm bounds to all second-order derivatives, a scalar upper bound on the remainder term can be expressed as:

$$\Big\| \tfrac{1}{2} \sum_{p,q} \langle H_{pq} \delta h_p, \delta h_q \rangle \Big\| \leq \tfrac{M_\Phi}{2} \Big( \sum_{j \in \mathcal{S}} \|\delta\boldsymbol{h}_j\| \Big)^2, \tag{22}$$

where $M_\Phi$ depending only on $M$ and $L$. Accordingly, the final editing performance can be decomposed as

$$E_r(S) = \hat{\boldsymbol{h}}_L^r - \boldsymbol{h}_L^r = \sum_{j \in \mathcal{S}} J_{j \to L} \delta\boldsymbol{h}_j + R, \qquad \|R\| \leq \tfrac{C}{2} \Big( \sum_{j \in \mathcal{S}} \|\delta\boldsymbol{h}_j\| \Big)^2, \tag{23}$$

where $C = M_\Phi$. We define the per-layer editing performance as $e_j^r := J_{j \to L}\,\delta\boldsymbol{h}_j$, so that the total editing performance of employing representation editing can be expressed as:

$$E_r(S) \approx \sum_{j \in \mathcal{S}} \boldsymbol{e}_j^r, \tag{24}$$

up to the quadratic remainder $R$.

*Model editing.* Consider small modifications $\Delta\boldsymbol{W}_j$ to the weights $\boldsymbol{W}_j$ of layers $j \in \mathcal{S}$. Under our $C^2$ assumptions, the map $\boldsymbol{W}_j \mapsto \boldsymbol{h}_j$ satisfies a second-order Taylor expansion:

$$\boldsymbol{h}_j(\boldsymbol{W}_j + \Delta\boldsymbol{W}_j) = \boldsymbol{h}_j(\boldsymbol{W}_j) + D_{\boldsymbol{W}_j}\boldsymbol{h}_j(\boldsymbol{W}_j)\,\mathrm{vec}(\Delta\boldsymbol{W}_j) + O\big(\|\mathrm{vec}(\Delta\boldsymbol{W}_j)\|^2\big), \tag{25}$$

where $D_{\boldsymbol{W}_j}\boldsymbol{h}_j$ is the partial Fréchet derivative of $\boldsymbol{h}_j$ with respect to $\boldsymbol{W}_j$. Composing this with $G_j$ and applying the chain rule, the first-order contribution can be expressed as

$$D_{\boldsymbol{h}_j} G_j(\boldsymbol{h}_j)\,D_{\boldsymbol{W}_j}\boldsymbol{h}_j(\boldsymbol{W}_j)\,\mathrm{vec}(\Delta\boldsymbol{W}_j) = J_{j \to L}^W\,\mathrm{vec}(\Delta\boldsymbol{W}_j), \tag{26}$$

where $J_{j \to L}^W$ is the Jacobian mapping weight perturbations at layer $j$ to the final representation. By applying operator-norm bounds to all second-order derivatives, the scalar upper bound on the remainder term can be expressed as:

$$\left\| R' \right\| \leq \frac{C'}{2} \Big( \sum_{j \in \mathcal{S}} \|\mathrm{vec}(\Delta \boldsymbol{W}_j)\| \Big)^2, \tag{27}$$

where $C'$ depends only on the layer count $L$ and the per-layer second-derivative bounds. Thus, the total editing performance of model editing can be decomposed as:

$$E_m(\mathcal{S}) = \hat{\boldsymbol{h}}_L^m - \boldsymbol{h}_L = \sum_{j \in \mathcal{S}} J_{j \to L}^W \mathrm{vec}(\Delta \boldsymbol{W}_j) + R'. \tag{28}$$

Similarly, let the per-layer editing performance $\boldsymbol{e}_j^m := J_{j \to L}^W \mathrm{vec}(\Delta \boldsymbol{W}_j)$. Then, the overall editing performance of model editing across the selected layers $\mathcal{S}$ can be expressed as:

$$E_r(\mathcal{S}) \approx \sum_{j \in \mathcal{S}} \boldsymbol{e}_j^m, \tag{29}$$

up to the quadratic remainder $R'$.

**Remark.** When the per-layer editing is sufficiently small, the first-order terms dominate, and the contributions from different layers can be approximated as linearly additive, yielding $E(\mathcal{S}) \approx \sum_{j \in \mathcal{S}} e_j$. However, as the editing magnitudes increase, the quadratic remainder terms $R$ and $R'$ become non-negligible, leading to higher-order interactions between layers. Such interactions can diminish the overall editing performance, consistent with experimental results showing that excessive layer edits reduce evasion performance.

## C.2 SINGEL LAYER EDITING PERFORMANCE COMPARISON RATIO.

Consider a pre-norm transformer-based LLM with $L$ layers, let $\{r_t\}_{1 \leq t \leq L}$ denote the residual increments at layer $t$. We assume that:

1. $\mathbb{E}[r_t] = 0$ for all $t$,
2. $\mathbb{E}\|r_t\|_2^2 = \sigma^2$ and $\sup_t \mathbb{E}\|r_t\|_2^2 < \infty$,
3. cross-covariances are uniformly bounded so that the total contribution of cross terms grows at most linearly: $\sum_{s \neq t} |\mathbb{E}\langle r_s, r_t \rangle| = o(j^2)$ and in particular $\sum_{s \neq t} |\mathbb{E}\langle r_s, r_t \rangle| = O(j)$.

$\boldsymbol{P}_j$ denotes the projection matrix onto the difference space between AIGT and HWT at layer $j$. By employing model editing, the resulting hidden representation $\hat{\boldsymbol{h}}_j^m$ at layer $j$ can be expressed as:

$$\hat{\boldsymbol{h}}_j^m = \boldsymbol{h}_j' + (I - \boldsymbol{P}_j)\boldsymbol{W}_{2,j}\boldsymbol{z}_j, \tag{30}$$

where $\boldsymbol{h}_j' = \boldsymbol{h}_{j-1}' + FFN(\boldsymbol{h}_{j-1}')$ and $\boldsymbol{z}_j = \sigma(\boldsymbol{W}_1 \boldsymbol{h}_{j-1}')$. In contrast, the resulting hidden representation $\hat{\boldsymbol{h}}_j^r$ at layer $j$ after employing R-EAT can be expressed as

$$\hat{\boldsymbol{h}}_j^r = (I - \boldsymbol{P}_j)\boldsymbol{h}_j. \tag{31}$$

We define the per-layer editing performance ratio $R_j$ between R-EAT and model editing as:

$$R := \frac{\|\boldsymbol{e}_j^r\|_F}{\|\boldsymbol{e}_j^m\|_F} \approx \frac{\|\hat{\boldsymbol{h}}_j^r - \boldsymbol{h}_j\|_F}{\|\hat{\boldsymbol{h}}_j^m - \boldsymbol{h}_j\|_F} = \frac{\|\boldsymbol{P}(\boldsymbol{h}_j' + \boldsymbol{W}_{2,j}\boldsymbol{z}_j)\|_F}{\|\boldsymbol{P}\boldsymbol{W}_{2,j}\boldsymbol{z}_j\|_F} \propto \sqrt{j}, \tag{32}$$

where $\| \cdot \|_F$ denotes the Frobenius norm.

**Proof.** As described in Appendix C.1, the single-layer editing performance of R-EAT and model editing at layer $j$ are defined as

$$\boldsymbol{e}_j^r := J_{j \to L}\, \delta \boldsymbol{h}_j, \qquad \boldsymbol{e}_j^m := J_{j \to L}^W \mathrm{vec}(\Delta \boldsymbol{W}_j), \tag{33}$$

where $J_{j \to L}$ and $J_{j \to L}^W$ denote the Jacobians mapping local perturbations to the final hidden state.

To analyze the effect of a small modification $\delta\boldsymbol{h}_j$ at layer $j$, consider the first-order Taylor expansion of the downstream map $G_j = F_L \circ \cdots \circ F_{j+1}$ around $\boldsymbol{h}_j$:

$$G_j(\boldsymbol{h}_j + \delta\boldsymbol{h}_j) = G_j(\boldsymbol{h}_j) + D_{\boldsymbol{h}_j} G_j(\boldsymbol{h}_j)\,\delta\boldsymbol{h}_j + R_j, \tag{34}$$

where the remainder satisfies $\|R_j\| \leq \frac{M}{2}\|\delta\boldsymbol{h}_j\|^2$.

For R-EAT, the modification is given by $\delta\boldsymbol{h}_j^r = \hat{\boldsymbol{h}}_j^r - \boldsymbol{h}_j$, while for model editing it is $\delta\boldsymbol{h}_j^m = D_{\boldsymbol{W}_j}\boldsymbol{h}_j \operatorname{vec}(\Delta\boldsymbol{W}_j)$. Substituting these expressions yields

$$\boldsymbol{e}_j^r = J_{j \to L}\delta\boldsymbol{h}_j^r \approx \hat{\boldsymbol{h}}_j^r - \boldsymbol{h}_j, \qquad \boldsymbol{e}_j^m = J_{j \to L}^W \operatorname{vec}(\Delta\boldsymbol{W}_j) \approx \hat{\boldsymbol{h}}_j^m - \boldsymbol{h}_j, \tag{35}$$

where the approximations are justified by the fact that the quadratic remainder terms are of order $O(\|\delta\boldsymbol{h}_j\|^2)$ and hence negligible under small perturbations. Consequently, the per-layer editing performance ratio is

$$R := \frac{\|\boldsymbol{e}_j^r\|_F}{\|\boldsymbol{e}_j^m\|_F} \approx \frac{\|\hat{\boldsymbol{h}}_j^r - \boldsymbol{h}_j\|_F}{\|\hat{\boldsymbol{h}}_j^m - \boldsymbol{h}_j\|_F} = \frac{\|\boldsymbol{P}_j(\boldsymbol{h}_j' + \boldsymbol{W}_{2,j}\boldsymbol{z}_j)\|_F}{\|\boldsymbol{P}_j\boldsymbol{W}_{2,j}\boldsymbol{z}_j\|_F}. \tag{36}$$

Under the pre-norm architecture, $\boldsymbol{h}_j'$ can be expressed as:

$$\boldsymbol{h}_j' \approx \boldsymbol{h}_0 + \sum_{t=1}^{j-1} r_t, \tag{37}$$

where $r_t$ denotes the net increment contributed by the layer $t$ (the composite effect of $MHA$ and $FFN$ after layer-norm). Consider the squared Frobenius norm of the projected numerator:

$$\mathbb{E}\big\|\boldsymbol{P}_j(\boldsymbol{h}_j' + \boldsymbol{W}_{2,j}\boldsymbol{z}_j)\big\|_2^2 = \mathbb{E}\Big\|\boldsymbol{P}_j\Big(\boldsymbol{h}_0 + \sum_{t=1}^{j-1} r_t + \boldsymbol{W}_{2,j}\boldsymbol{z}_j\Big)\Big\|_2^2$$

$$= \mathbb{E}\Big\|\boldsymbol{P}_j\Big(\sum_{t=1}^{j-1} r_t\Big)\Big\|_2^2 + 2\mathbb{E}\Big\langle\boldsymbol{P}_j\sum_{t=1}^{j-1} r_t, \boldsymbol{P}_j(\boldsymbol{h}_0 + \boldsymbol{W}_{2,j}\boldsymbol{z}_j)\Big\rangle + \|\boldsymbol{P}_j(\boldsymbol{h}_0 + \boldsymbol{W}_{2,j}\boldsymbol{z}_j)\|_2^2. \tag{38}$$

By the zero-mean assumption and bounded cross-covariance assumption the cross term is lower-order (at most $O(j)$) compared to the main variance term. Expanding the first term gives

$$\mathbb{E}\Big\|\boldsymbol{P}_j\Big(\sum_{t=1}^{j-1} r_t\Big)\Big\|_2^2 = \sum_{t=1}^{j-1}\mathbb{E}\|\boldsymbol{P}_j r_t\|_2^2 + 2\sum_{1 \leq s < t \leq j-1}\mathbb{E}\langle\boldsymbol{P}_j r_s, \boldsymbol{P}_j r_t\rangle. \tag{39}$$

By the zero-mean and bounded cross-covariance assumptions, the double sum of cross terms $2\sum_{1 \leq s < t \leq j-1}\mathbb{E}\langle\boldsymbol{P}_j r_s, \boldsymbol{P}_j r_t\rangle$ is of lower order compared to the main variance term $\sum_{t=1}^{j-1}\mathbb{E}\|\boldsymbol{P}_j r_t\|_2^2$, and can be treated as at most $O(j)$. Hence there exists constants $a, b > 0$ (independent of $j$) such that for all $j$:

$$\mathbb{E}\big\|\boldsymbol{P}_j(\boldsymbol{h}_j' + \boldsymbol{W}_{2,j}\boldsymbol{z}_j)\big\|_2^2 = a\,j + b + o(j). \tag{40}$$

By using Jensen's inequality, we obtain:

$$\mathbb{E}\big\|\boldsymbol{P}_j(\boldsymbol{h}_j' + \boldsymbol{W}_{2,j}\boldsymbol{z}_j)\big\|_2 \leq \sqrt{\mathbb{E}\big\|\boldsymbol{P}_j(\boldsymbol{h}_j' + \boldsymbol{W}_{2,j}\boldsymbol{z}_j)\big\|_2^2} \propto \sqrt{j}. \tag{41}$$

The denominator $\|\boldsymbol{P}_j\boldsymbol{W}_{2,j}\boldsymbol{z}_j\|_2$ depends on the single-layer FFN output $\boldsymbol{W}_{2,j}\boldsymbol{z}_j$. Under standard assumptions on weight regularity and layer normalization, this term remains stable and does not scale with the layer index $j$. Thus

$$\mathbb{E}\|\boldsymbol{P}_j\boldsymbol{W}_{2,j}\boldsymbol{z}_j\|_2^2 = c, \tag{42}$$

where constant $c > 0$ (independent of $j$).

By combining $\mathbb{E}\big\|\boldsymbol{P}_j(\boldsymbol{h}_j' + \boldsymbol{W}_{2,j}\boldsymbol{z}_j)\big\|_2^2$ and $\mathbb{E}\|\boldsymbol{P}_j\boldsymbol{W}_{2,j}\boldsymbol{z}_j\|_2^2$, we obtain:

$$\frac{\mathbb{E}\|\boldsymbol{P}_j(\boldsymbol{h}_j' + \boldsymbol{W}_{2,j}\boldsymbol{z}_j)\|_2}{\mathbb{E}\|\boldsymbol{P}_j\boldsymbol{W}_{2,j}\boldsymbol{z}_j\|_2} \propto \sqrt{j}. \tag{43}$$

Therefore, the per-layer performance ratio $R$ can be approximated as:

$$R \approx \frac{\|\boldsymbol{P}_j(\boldsymbol{h}_j' + \boldsymbol{W}_{2,j}\boldsymbol{z}_j)\|_F}{\|\boldsymbol{P}_j\boldsymbol{W}_{2,j}\boldsymbol{z}_j\|_F} \propto \sqrt{j} \tag{44}$$

# D   DETAILS OF DETECTORS

This section describes the experimental implementation settings for all detectors. We adopt GPT-J-6b (Wang & Komatsuzaki, 2021) as the default model for scoring in all statistics-based detectors.

**RoBERTa-base** and **RoBERTa-large**. This approach frames AI text detection as a binary classification problem. We utilized two models from the `RoBERTa-openai-detector` series: a `base` version and a `large` version. Both detectors are built upon the RoBERTa architecture (Liu et al., 2019) and have been specifically fine-tuned by OpenAI for the task of distinguishing between HWTs and AIGTs.

**Likelihood.** The average log probability of all tokens in the candidate text is used as the metric.

**Entropy.** The average entropy of each token is calculated based on the probability distribution over the vocabulary.

**DetectLRR.** The ratio of log-likelihood to log-rank is used as the average metric.

**Binoculars** (Hans et al., 2024a). The core intuition of this detector is that AIGT is typically more predictable (i.e., has a lower perplexity or higher average log-likelihood) under an instruction-tuned model (the observer) compared to its base pre-trained version (the performer). The detection score is calculated as the difference between the text's average log-likelihood under the observer and the performer. A large positive discrepancy suggests that the text aligns too closely with the patterns of the instruction-tuned model, marking it as likely machine-generated. This method tests for intrinsic statistical properties of the text rather than learned classification boundaries. In our experiments, we use `falcon-7b-instruct` as the observer and its base counterpart, `falcon-7b`, as the performer (Almazrouei et al., 2023).

**Lastde** (Li et al., 2024). It computes a detection score by dividing the Log-Likelihood of the text (global feature) by the aggregated MDE statistic (local feature), thus capturing both global and local statistical properties of token probability sequences.

**Lastde++** (Li et al., 2024). It enhances Lastde by incorporating fast-sampling–based normalization, which adjusts the score through the discrepancy between the observed Lastde value and its sampling distribution, yielding stronger discriminative power.

# E   EXPERIMENTAL DETAILS OF BASELINES

In this section, we provide a detailed description of the methodologies and hyperparameter configurations used to establish the experimental baselines.

**(1) Iterative DPO Fine-Tuning** (Pedrotti et al., 2024). This approach employs an iterative fine-tuning strategy using DPO (Rafailov et al., 2023a). The core idea is to progressively align the model's writing style with that of HWT. The process consists of two full iterations. In each iteration, a preference dataset is constructed by pairing HWT samples (as "chosen" responses) with machine-generated text (MGT) from the model being tuned (as "rejected" responses).

- First Iteration **(DPO-1):** The base model generates MGT. This MGT is paired with HWT to form the first preference dataset. The base model is then fine-tuned on this dataset using DPO to produce a first-iteration-tuned model.
- Second Iteration **(DPO-2):** The model from the first iteration is used to generate a new set of MGT. This new MGT is again paired with HWT to create a second preference dataset. The first-iteration-tuned model is further fine-tuned on this second data set, resulting in a second-iteration-tuned model. This allows for continuous refinement towards the human stylistic target.

The following hyperparameters were used for DPO fine-tuning, utilizing Low-Rank Adaptation (LoRA) (Hu et al., 2022a) for memory efficiency.

- **DPO & LoRA Configuration:**

    ```
    LoraConfig(
    ```

```
        r=32,
        lora_alpha=16,
        lora_dropout=0.05,
        target_modules=[
            "q_proj", "k_proj", "v_proj", "o_proj",
            "gate_proj", "up_proj", "down_proj",
        ],
        bias="none",
        task_type="CAUSAL_LM"
    )
```

- **Training Hyperparameters:**

```
TrainingArguments(
    learning_rate=5e-6,
    num_train_epochs=1,
    per_device_train_batch_size=1,
    gradient_accumulation_steps=4,
    optim="paged_adamw_8bit",
    beta=0.1,
    warmup_ratio=0.1
)
```

- **Text Generation Parameters:**

```
Max Tokens: 512
Temperature: 1.0
Top-p: 1.0
```

**(2) HUMPA** (Wang et al., 2025). The method consists of two main stages: it first fine-tuned a small language model (SLM) using DPO as a proxy. During text generation, the proxy model is used to adjust the logits of the target LLM, guiding it to produce more human-like text. In our experiments, we use `Llama-2-7b-chat-hf` (Touvron et al., 2023) and a `Qwen-3 8B` (Team, 2024) as the SLM.

The following parameters were used to create the reference model and execute the HUMPA attack.

- **DPO & LoRA for the reference model:** The reference model was created by fine-tuning a base model using DPO with 4-bit NF4 quantization.

```
LoraConfig(
    r=16,
    lora_alpha=32,
    lora_dropout=0.05,
    target_modules=[
        "q_proj", "k_proj", "v_proj", "o_proj",
        "gate_proj", "up_proj", "down_proj",
    ],
    bias="none",
    task_type="CAUSAL_LM"
)
```

- **Training for a Reference Model:**

```
TrainingArguments(
    learning_rate=5e-5,
    num_train_epochs=3,
    per_device_train_batch_size=1,
    gradient_accumulation_steps=8,
```

```
        optim="paged_adamw_8bit",
        beta=0.1,
        warmup_ratio=0.03
    )
```

- **Steered Text Generation:**

```
    Steering Strength (alpha): 0.8
    Max New Tokens: 150
    Temperature: 1.0
    Top-p: 1.0
    Do Sample: True
```

**(3) ModelEditing.** This method modifies model weights by constructing a difference space between preferred and non-preferred texts, providing a training-free alternative to DPO. Following the original experimental setup, we perform editing starting from the intermediate layers. In all experiments, editing begins at layer 20 with an editing strength of 1.

# F  ADDITIONAL EXPERIMENTAL RESULTS

## F.1  EXPERIMENTAL RESULTS ON ADDITIONAL CORPORA

We incorporate two new corpora: PubMedQA (Jin et al., 2019) for biomedical research question answering and SQuAD (Rajpurkar et al., 2016) for Wikipedia contexts. For PubMedQA, we only used the questions as prompts to guide the LLM in text generation. All experimental settings remain consistent with those in the original manuscript. The average AUROC and AUPRC results for Llama2-13b on the two new corpora are provided in Table 3 and 4. Experimental results demonstrate that our R-EAT achieves optimal performance on both the PubMedQA and SQuAD datasets, reducing the average AUROC of 12 detectors to 57.53% and 59.07%, respectively.

## F.2  EXPERIMENTAL RESULTS ON ADDITIONAL DETECTORS

To further validate our experimental results, we introduce four additional detectors: **(1) Radar** (Hu et al., 2023) is a robust detector based on adversarial learning. **(2) ImBD** (Chen et al., 2025) optimizes a scoring LLM through style preference to mimic the machine's preferred text style distribution, and then uses this model to compute Style Conditional Probability Curvature (Style-CPC), which efficiently detects machine-modified text. **(3) Text-Fluoroscopy** (Yu et al., 2024) identifies the largest distributional differences in embeddings projected into the vocabulary space from intermediate layers of the LLM (instead of the first or last layers), capturing the intrinsic features of the text to achieve better generalization and superior detection performance for AIGT. **(4) Fast-detectGPT** (Bao et al., 2023) utilizes conditional probability curvature to elucidate discrepancies in word choices between LLMs and humans within a given context. In this study, both the sampling model and the scoring model used in the method are gpt-j-6b (Wang & Komatsuzaki, 2021). As Shown in Table 5, R-EAT achieves the lowest average AUROC and average AUPRC on OpenWebText, resulting in the final evasion detection performance. On the WritingPrompt dataset, our R-EAT outperforms HUMPA and model editing, performing similarly to DPO-1 and DPO-2, which are based on direct fine-tuning.

## F.3  THE GENERALIZATION OF DIFFERENCE SPACE.

We investigate the generalization of the difference space on Llama-13b. As shown in the Table 6, R-EAT builds a difference space from the OpenWebText dataset. When applied to WritingPrompt prompts, it achieves evasion detection performance that even surpasses the baseline trained directly

Table 3: AUROC results of different evasion detection methods on PubMedQA and SQuAD using Llama2-13b.The best results are highlighted in bold, and the second-best results are marked with underline.

|  | | Base | HUMPA | DPO-1 | DPO-2 | ModelEditing | R-EAT (ours) |
|---|---|---|---|---|---|---|---|
| | RoBERTa-base | 0.7544 | **0.6625** | 0.7555 | 0.7231 | 0.8605 | 0.6688 |
| | RoBERTa-large | 0.7604 | 0.6878 | 0.6866 | **0.6861** | 0.8098 | 0.7006 |
| | Likelihood | 0.9989 | 0.9449 | 0.9919 | 0.9934 | **0.8496** | 0.8667 |
| | Entorpy | **0.0231** | 0.1049 | 0.0271 | 0.0252 | 0.0894 | 0.0574 |
| | DetectLRR | 0.9945 | **0.8751** | 0.9736 | 0.9713 | 0.9791 | 0.8952 |
| | Binoculars | 0.8562 | 0.7970 | 0.9308 | 0.9258 | 0.8289 | **0.7110** |
| PubMedQA | Lastde | 0.9403 | 0.6186 | 0.7940 | 0.7760 | 0.6545 | **0.3351** |
| | Lasede++ | 0.9967 | 0.8533 | 0.9937 | 0.9888 | 0.8496 | **0.3707** |
| | Fast-detectGPT | 0.9967 | **0.8663** | 0.9953 | 0.9917 | 0.9955 | 0.9954 |
| | ImBD | 0.9763 | 0.7762 | 0.9781 | 0.9626 | 0.7831 | **0.2712** |
| | Radar | 0.2864 | 0.2811 | 0.1884 | **0.1870** | 0.4853 | 0.3344 |
| | Text-Fluoroscopy | 0.9062 | 0.7728 | 0.9473 | 0.9394 | **0.6532** | 0.6704 |
| | Average | 0.7908 | 0.6867 | 0.7719 | 0.7642 | 0.7365 | **0.5731** |
| | RoBERTa-base | 0.9608 | 0.8971 | 0.9365 | 0.9227 | 0.9341 | **0.5766** |
| | RoBERTa-large | 0.9540 | 0.8964 | 0.9295 | 0.9232 | 0.9296 | **0.5787** |
| | Likelihood | 0.9507 | 0.7960 | 0.9368 | 0.9176 | 0.9133 | **0.7530** |
| | Entorpy | 0.3194 | 0.4637 | 0.2120 | 0.2251 | 0.2440 | **0.1287** |
| | DetectLRR | 0.9744 | 0.8957 | 0.9492 | 0.9327 | 0.9367 | **0.8549** |
| | Binoculars | 0.9279 | 0.8596 | 0.9518 | 0.9470 | 0.8933 | **0.6600** |
| SQuAD | Lastde | 0.9508 | 0.8836 | 0.9036 | 0.8777 | 0.8578 | **0.5589** |
| | Lasede++ | 0.9729 | 0.9012 | 0.9773 | 0.9594 | 0.8572 | **0.4075** |
| | Fast-detectGPT | 0.9784 | **0.9137** | 0.9843 | 0.9664 | 0.9864 | 0.9864 |
| | ImBD | 0.9848 | 0.9284 | 0.9859 | 0.9734 | 0.9810 | **0.4280** |
| | Radar | 0.8245 | 0.7720 | 0.7769 | 0.7579 | 0.8311 | **0.6581** |
| | Text-Fluoroscopy | 0.9165 | 0.8440 | 0.9695 | 0.9500 | **0.5378** | 0.5565 |
| | Average | 0.8929 | 0.8376 | 0.8761 | 0.8628 | 0.8252 | **0.5907** |

on WritingPrompt. Its performance is only slightly lower than using WritingPrompt itself to construct the space. These results show that R-EAT has strong generalization ability.

### F.4 EXPERIMENTAL RESULTS USING QWEN3-14B.

We conduct experiments on Qwen3-14b using the same hyperparameter settings as those for Llama2-13b. As shown in the Table 7, R-EAT effectively evades eight detectors, reducing the average detection accuracy to 18.89% and 82.62% on the two datasets, respectively, outperforming HUMPA and model-editing-based methods. At the same time, we observe that R-EAT performs slightly worse than DPO-1 and DPO-2 on both datasets. Pervious ablation studies in section 4.7 indicate that selecting more appropriate hyperparameters can further enhance R-EAT's performance. Notably, R-EAT substantially reduces the computational resources required compared to fine-tuning-based approaches.

### F.5 COMPARISON OF AURPC ACROSS DIFFERENT DETECTORS.

In this section, we evaluate different evasion methods on Llama2-13b and Qwen3-14b using AUPRC, considering eight detectors across two datasets. As shown in Table 8, R-EAT achieves state-of-the-art performance on Llama2-13b, reducing AUPRC by 18.65% and 14.93% on the two datasets, respectively. Table 9 illustrates the AUPRC of different evasion methods on Qwen3-14b. With the same parameter settings as in Llama2-13b, R-EAT attains lower AUPRC than HUMPA and model editing, while its performance is more comparable to that of direct fine-tuning. We believe that a better choice of hyperparameters can further improve the evasion performance of R-EAT.

Table 4: AUPRC results of different evasion detection methods on PubMedQA and SQuAD using Llama2-13b.The best results are highlighted in bold, and the second-best results are marked with underline.

|  |  | Base | HUMPA | DPO-1 | DPO-2 | ModelEditing | R-EAT (ours) |
|---|---|---|---|---|---|---|---|
| PQAU | RoBERTa-base | 0.8098 | **0.7302** | 0.8394 | 0.8168 | 0.9000 | **0.7575** |
|  | RoBERTa-large | 0.7876 | 0.7333 | 0.7379 | **0.7296** | 0.8381 | 0.7430 |
|  | Likelihood | 0.9990 | 0.9517 | 0.9942 | 0.9942 | 0.8903 | **0.8796** |
|  | Entorpy | 0.3089 | 0.3199 | 0.3089 | **0.3086** | 0.3172 | 0.3120 |
|  | DetectLRR | 0.9953 | 0.8887 | 0.9783 | 0.9757 | 0.9768 | **0.8845** |
|  | Binoculars | 0.8482 | 0.7787 | 0.9045 | 0.9013 | 0.8085 | **0.6765** |
|  | Lastde | 0.9350 | 0.6417 | 0.7711 | 0.7616 | 0.7020 | **0.4875** |
|  | Lasede++ | 0.9977 | 0.8936 | 0.9953 | 0.9924 | 0.8903 | **0.5053** |
|  | Fast-detectGPT | 0.9978 | **0.9030** | 0.9966 | 0.9941 | 0.9967 | 0.9967 |
|  | ImBD | 0.9793 | 0.8004 | 0.9822 | 0.9753 | 0.7858 | **0.3842** |
|  | Radar | 0.9337 | 0.8197 | 0.9620 | 0.9554 | **0.7073** | 0.7259 |
|  | Text-Fluoroscopy | 0.3709 | **0.3739** | 0.3392 | 0.3390 | 0.4981 | 0.4073 |
|  | Average | 0.8303 | 0.7362 | 0.8175 | 0.8120 | 0.7759 | **0.6467** |
| SQuAD | RoBERTa-base | 0.9658 | 0.9119 | 0.9524 | 0.9462 | 0.9453 | **0.6439** |
|  | RoBERTa-large | 0.9501 | 0.8985 | 0.9344 | 0.9321 | 0.9215 | **0.5749** |
|  | Likelihood | 0.9437 | **0.7861** | 0.9455 | 0.9341 | 0.9267 | 0.7906 |
|  | Entorpy | 0.3876 | 0.4714 | 0.3517 | 0.3720 | 0.3740 | **0.3288** |
|  | DetectLRR | 0.9813 | 0.9121 | 0.9634 | 0.9550 | 0.9556 | **0.8673** |
|  | Binoculars | 0.9403 | 0.8665 | 0.9547 | 0.9474 | 0.8995 | **0.7071** |
|  | Lastde | 0.9608 | 0.8794 | 0.9168 | 0.8965 | 0.8907 | **0.6753** |
|  | Lasede++ | 0.9822 | 0.9160 | 0.9830 | 0.9735 | 0.8877 | **0.4864** |
|  | Fast-detectGPT | 0.9861 | **0.9298** | 0.9885 | 0.9779 | 0.9909 | 0.9909 |
|  | ImBD | 0.9891 | 0.9397 | 0.9886 | 0.9767 | 0.9821 | **0.5107** |
|  | Radar | 0.9374 | 0.8838 | 0.9768 | 0.9653 | 0.6390 | **0.6375** |
|  | Text-Fluoroscopy | 0.7185 | 0.6792 | 0.6761 | 0.6604 | 0.7707 | **0.6064** |
|  | Average | 0.8952 | 0.8395 | 0.8860 | 0.8781 | 0.8486 | **0.6517** |

### F.6 IMPACT OF EDITING STRENGTH $\alpha$.

We calculate the $|\Delta PPL|$ of the generated texts at different editing strengths to evaluate the impact of editing strength on text quality. As shown in Figure 11, when the editing strength exceeds a certain threshold (e.g., $\alpha = 0.7$), the semantic integrity of the generated text is compromised, leading to a decline in performance.

### F.7 IMPACT OF THE NUMBER OF SAMPLES.

We conduct a detailed investigation by extracting the number of singular vectors required for constructing the difference space at different sample sizes (refer to 'Dimension'), as well as calculating the cosine of the principal angle between the difference space constructed with multiple samples and the one built with 100 samples (refer to 'Cosine Value').

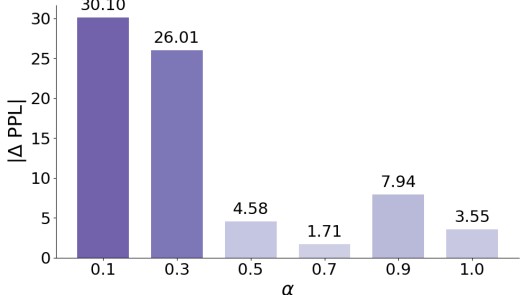

Figure 11: Impact of Different Editing Strengths on Text Quality.

Table 5: Evaluation results of different evasion detection methods against new detectors using Llama2-13b.

| | Fast-detectGPT | | ImBD | | Text Fluoroscopy | | Radar | | Average | |
|---|---|---|---|---|---|---|---|---|---|---|
| | auroc | auprc | auroc | auprc | auroc | auprc | auroc | auprc | auroc | auprc |
| OpenWebText | | | | | | | | | | |
| Base | 0.9808 | 0.9755 | 0.9549 | 0.9580 | 0.9650 | 0.9705 | 0.9886 | 0.9892 | 0.9723 | 0.9733 |
| HUMPA | 0.7249 | 0.7339 | 0.4357 | 0.4355 | 0.8194 | 0.8673 | 0.9993 | 0.9993 | 0.7448 | 0.7590 |
| DPO-1 | 0.6814 | 0.7775 | 0.5766 | 0.6623 | 0.8753 | 0.9007 | **0.9080** | **0.9150** | 0.7603 | 0.8139 |
| DPO-2 | 0.6587 | 0.7568 | 0.5619 | 0.6412 | 0.8630 | 0.8909 | 0.9109 | 0.9183 | 0.7486 | 0.8018 |
| ModelEditing | 0.9943 | 0.9966 | 0.9686 | 0.9717 | 0.7731 | 0.8213 | 0.9809 | 0.9834 | 0.9292 | 0.9433 |
| R-EAT (Ours) | **0.3379** | **0.4501** | **0.3062** | **0.4103** | **0.6223** | **0.6885** | 0.9408 | 0.9530 | **0.5518** | **0.6255** |
| WritingPrompt | | | | | | | | | | |
| Base | 0.9932 | 0.9958 | 0.9943 | 0.9955 | 0.9680 | 0.9780 | 0.9502 | 0.9467 | 0.9764 | 0.9790 |
| HUMPA | 0.9698 | 0.9764 | 0.9453 | 0.9427 | 0.8793 | 0.9036 | 0.9728 | 0.9748 | 0.9418 | 0.9494 |
| DPO-1 | 0.6593 | 0.7790 | **0.6798** | **0.7580** | 0.7527 | 0.8089 | **0.8103** | **0.8372** | 0.7255 | **0.7958** |
| DPO-2 | 0.6392 | 0.7664 | 0.6905 | 0.7685 | 0.7390 | 0.8024 | 0.8243 | 0.8533 | **0.7233** | 0.7977 |
| ModelEditing | 0.9834 | 0.9775 | 0.9949 | 0.9961 | 0.7880 | 0.8387 | 0.9541 | 0.9568 | 0.9301 | 0.9423 |
| R-EAT (Ours) | **0.6294** | **0.7015** | 0.7635 | 0.7478 | **0.6951** | **0.7741** | 0.9007 | 0.9146 | 0.7472 | 0.7845 |

Table 6: comparison of AUROC on WritingPrompt using Llama2-13b. R-EAT$_{ood}$ denotes constructing the difference space with the OpenWebText dataset, while R-EAT$_{id}$ denotes constructing the space with the WritingPrompt dataset. The best results are highlighted in bold.

| Detector | Base | HUMPA | DPO-1 | DPO-2 | ModelEditing | R-EAT$_{id}$ | R-EAT$_{ood}$ |
|---|---|---|---|---|---|---|---|
| RoBERTa-base | 0.9715 | 0.9662 | 0.8558 | 0.8680 | 0.9873 | **0.6991** | 0.8100 |
| RoBERTa-large | 0.9437 | 0.9246 | 0.8918 | 0.8757 | 0.9524 | **0.6604** | 0.7753 |
| Likelihood | 0.9843 | 0.8468 | **0.5536** | 0.5578 | 0.9694 | 0.8326 | 0.8393 |
| Entropy | 0.1260 | 0.5336 | 0.5534 | 0.5317 | 0.3615 | 0.1613 | **0.1503** |
| DetectLRR | 0.9864 | 0.9096 | **0.5703** | 0.5914 | 0.9775 | 0.8314 | 0.8899 |
| Binoculars | 0.9389 | 0.9017 | 0.8673 | 0.8610 | 0.9102 | **0.7376** | 0.7882 |
| Lastde | 0.9990 | 0.9921 | **0.7042** | 0.7188 | 0.9928 | 0.8977 | 0.8694 |
| Lastde++ | 0.9905 | 0.9637 | 0.6399 | **0.6110** | 0.9846 | 0.6827 | 0.6582 |
| Average | 0.8675 | 0.8798 | 0.7045 | 0.7019 | 0.8920 | **0.6879** | 0.7004 |

As shown in Table 10, we find that as the sample size increases, the number of singular vectors satisfying $\tau > 90\%$ also increases. This indicates a significant rise in the dimensionality and complexity of the difference space. Additionally, we observe that the cosine of the principal angle between the difference space constructed with 200 samples and the one built with 100 samples is the lowest. This suggests that the difference space at this point introduces a significant amount of secondary features and sample-specific noise, leading to a substantial decline in evasion performance. When the sample size continues to grow, although the direction of the difference space begins to revert toward the primary direction, the overall dimensionality of the difference space continues to increase, causing the proportion and absolute amount of noisy features to rise. Consequently, the evasion performance starts to recover, but it still remains below the optimal level achieved with 100 samples.

Table 10: Analysis of Difference Spaces Constructed with Different Sample Sizes.

| sample numbers | Dimension | Cosine Value |
|---|---|---|
| 100 | 10 | - |
| 200 | 125 | 0.1884 |
| 300 | 180 | 0.2087 |
| 400 | 236 | 0.2465 |
| 500 | 285 | 0.2739 |

Table 7: Comparison of AUROC across different detectors on OpenWebText and WritingPrompt dataset using Qwen3-14b. The best results are highlighted in bold.

| Datasets | Detector | Base | HUMPA | DPO-1 | DPO-2 | ModelEditing | R-EAT |
|---|---|---|---|---|---|---|---|
| OpenWebText | RoBERTa-base | 0.9715 | 0.9622 | 0.8077 | **0.7935** | 0.9987 | 0.9049 |
| | RoBERTa-large | 0.9491 | 0.9320 | **0.8629** | 0.8665 | 0.9993 | 0.9007 |
| | Likelihood | 0.9477 | 0.6350 | 0.3649 | **0.3971** | 0.9382 | 0.9413 |
| | Entropy | 0.2934 | 0.8742 | 0.7473 | 0.7153 | 0.3983 | **0.1369** |
| | DetectLRR | 0.9943 | 0.9779 | **0.4956** | 0.5106 | 0.9811 | 0.7215 |
| | Binoculars | 0.8644 | **0.8484** | 0.9191 | 0.9277 | 0.9658 | 0.9122 |
| | Lastde | 0.9882 | 0.9779 | 0.6332 | 0.6333 | 0.9940 | **0.9321** |
| | Lastde++ | 0.9677 | 0.9296 | 0.7156 | **0.7056** | 0.9704 | 0.8614 |
| | Average | 0.8720 | 0.8921 | **0.6933** | 0.6937 | 0.9057 | 0.7889 |
| WritingPrompt | RoBERTa-base | 0.9599 | 0.9622 | 0.7897 | **0.7725** | 0.9996 | 0.9835 |
| | RoBERTa-large | 0.9348 | 0.9320 | 0.8434 | **0.8291** | 0.9994 | 0.9927 |
| | Likelihood | 0.9850 | 0.8722 | **0.7207** | 0.7362 | 0.9869 | 0.9876 |
| | Entropy | 0.2908 | 0.5313 | 0.4599 | 0.4476 | 0.1657 | **0.0170** |
| | DetectLRR | 0.9901 | 0.9599 | 0.7503 | 0.7663 | 0.9907 | **0.7154** |
| | Binoculars | 0.8803 | **0.8484** | 0.8653 | 0.8564 | 0.9236 | 0.8695 |
| | Lastde | 0.9976 | 0.9957 | **0.8070** | 0.8035 | 0.9985 | 0.9570 |
| | Lastde++ | 0.9902 | 0.9981 | 0.9169 | 0.9154 | 0.9964 | **0.8955** |
| | Average | 0.8786 | 0.8875 | 0.7692 | **0.7659** | 0.8826 | 0.8023 |

Table 8: Comparison of AUPRC across different detectors on OpenWebText and WritingPrompt dataset using llama2-13b. The best results are highlighted in bold.

| Datasets | Detector | Base | HUMPA | DPO-1 | DPO-2 | ModelEditing | R-EAT |
|---|---|---|---|---|---|---|---|
| OpenWebText | RoBERTa-base | 0.9805 | 0.9643 | 0.8890 | 0.8739 | 0.9760 | **0.8312** |
| | RoBERTa-large | 0.9718 | 0.9653 | 0.9074 | 0.9002 | 0.9720 | **0.8001** |
| | Likelihood | 0.9367 | **0.3152** | 0.6343 | 6426 | 0.8704 | 0.7430 |
| | Entropy | 0.4118 | 0.9662 | 0.6931 | 0.6636 | 0.5432 | **0.3998** |
| | DetectLRR | 0.9737 | **0.3740** | 0.7259 | 0.7335 | 0.9582 | 0.7738 |
| | Binoculars | 0.9180 | 0.8970 | 0.9276 | 0.9062 | 0.9439 | **0.8183** |
| | Lastde | 0.9725 | **0.5880** | 0.7746 | 0.8008 | 0.9623 | 0.8117 |
| | Lastde++ | 0.9739 | 0.6359 | 0.7498 | 0.7366 | 0.8893 | **0.4693** |
| | Average | 0.8924 | 0.7132 | 0.7877 | 0.7822 | 0.8894 | **0.7059** |
| WritingPrompt | RoBERTa-base | 0.9746 | 0.9679 | 0.8899 | 0.8959 | 0.9885 | **0.7495** |
| | RoBERTa-large | 0.9530 | 0.9363 | 0.9128 | 0.8994 | 0.9567 | **0.7139** |
| | Likelihood | 0.9823 | 0.8476 | **0.6778** | 0.6859 | 0.9661 | 0.8614 |
| | Entropy | **0.3291** | 0.5264 | 0.6539 | 0.6371 | 0.4116 | 0.3420 |
| | DetectLRR | 0.9922 | 0.9335 | **0.7135** | 0.7315 | 0.9854 | 0.8676 |
| | Binoculars | 0.9991 | 0.9201 | 0.8781 | 0.8777 | 0.9350 | **0.8053** |
| | Lastde | 0.9991 | 0.9934 | 0.8049 | 0.8161 | 0.9951 | **0.9235** |
| | Lastde++ | 0.9940 | 0.9702 | 0.7613 | 0.7442 | 0.9887 | **0.7226** |
| | Average | 0.8975 | 0.8869 | 0.7865 | 0.7860 | 0.9034 | **0.7482** |

## F.8 IMPACT OF EDITING LAYER.

We evaluate the impact of the number of edited layers on model performance under a small editing strength ($\alpha = 0.5$). The Figure 12 illustrates that when editing strength is small, single-layer editing is almost ineffective. Increasing the number of edited layers (e.g., 38–39 to 37–39) progressively enhances evasion performance, in line with our analysis in Section 3.3.

Table 9: Comparison of AUPRC across different detectors on OpenWebText and WritingPrompt dataset using Qwen3-14b. The best results are highlighted in bold.

| Datasets | Detector | Base | HUMPA | DPO-1 | DPO-2 | ModelEditing | R-EAT |
|---|---|---|---|---|---|---|---|
| OpenWebText | RoBERTa-base | 0.9746 | 0.9625 | 0.8360 | **0.8281** | 0.9989 | 0.9303 |
| | RoBERTa-large | 0.9546 | 0.9381 | 0.8804 | **0.8793** | 0.9993 | 0.9295 |
| | Likelihood | 0.9555 | 0.6061 | **0.4717** | 0.4865 | 0.9354 | 0.9518 |
| | Entropy | 0.4268 | 0.8914 | 0.7906 | 0.7630 | 0.5017 | **0.3414** |
| | DetectLRR | 0.9960 | 0.9652 | **0.5808** | 0.5984 | 0.9870 | 0.7985 |
| | Binoculars | 0.8848 | **0.8663** | 0.9307 | 0.9351 | 0.9651 | 0.9400 |
| | Lastde | 0.9824 | 0.9652 | 0.6248 | **0.6233** | 0.9868 | 0.9566 |
| | Lastde++ | 0.9668 | 0.9199 | 0.7567 | **0.7485** | 0.9663 | 0.8549 |
| | Average | 0.8927 | 0.8893 | 0.7340 | **0.7328** | 0.9176 | 0.8379 |
| WritingPrompt | RoBERTa-base | 0.9631 | 0.9625 | 0.8007 | **0.7921** | 0.9996 | 0.9841 |
| | RoBERTa-large | 0.9435 | 0.9381 | 0.8496 | 0.8443 | 0.9994 | 0.9923 |
| | Likelihood | 0.9791 | 0.8428 | **0.7573** | 0.7653 | 0.9869 | 0.9916 |
| | Entropy | 0.3716 | 0.5070 | 0.5283 | 0.5131 | 0.3384 | **0.3141** |
| | DetectLRR | 0.9933 | 0.9624 | 0.8103 | 0.8193 | 0.9931 | **0.7927** |
| | Binoculars | 0.9073 | 0.8663 | 0.8875 | 0.8811 | 0.9221 | 0.9154 |
| | Lastde | 0.9980 | 0.9959 | 0.8395 | 0.8387 | 0.9989 | 0.9750 |
| | Lastde++ | 0.9930 | 0.9830 | 0.9355 | 0.9363 | 0.9968 | **0.9000** |
| | Average | 0.8936 | 0.8823 | 0.8011 | **0.7988** | 0.9041 | 0.8582 |

Interestingly, we observe that increasing the number of edited layers to four (layers 36–39) leads to a decrease in detection-evasion performance. We attribute this phenomenon to the fact that the editing introduced at layer 36 are large, making the higher-order terms in the nonlinear transformations non-negligible. As a result, the effective contribution of the edits is attenuated or distorted, which can reduce the net gain despite the additional layer being edited. This observation is consistent with our theoretical analysis presented in Appendix C.1.

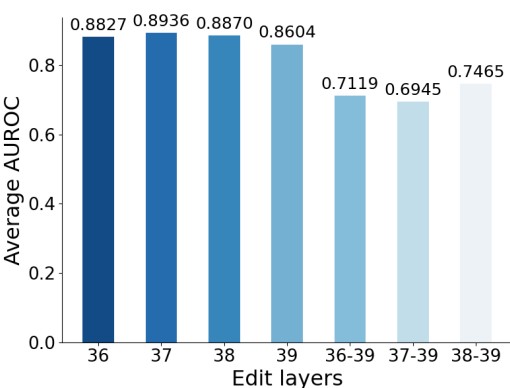

Figure 12: Impact of number of layers edited on $\alpha = 0.5$.

# G  POTENTIAL DEFENSE METHODS

We also suggest several potential strategies to defend against R-EAT. **(1) Adversarial training**, where detectors are retrained using AIGT after applying R-EAT. **(2) Ensemble detection**, which combines multiple detectors to make joint predictions.

