# OpenReview forum: "Toward Undetectable AI Text: AIGT detection evasion with representation editing"
_ICLR.cc/2026/Conference — ICLR 2026 Conference Desk Rejected Submission_

### Official Review · Reviewer_26kA · 2025-10-27

**Soundness:** 3
**Presentation:** 2
**Contribution:** 2
**Rating:** 6
**Confidence:** 4

**Summary:**

The paper introduces R-EAT (Representation Editing Attack), a training-free evasion method designed to help large language models (LLMs) generate text that is more difficult for AI-generated text (AIGT) detectors to recognize.

R-EAT constructs a “difference space” between human-written text (HWT) and AI-generated text (AIGT) by comparing their hidden representations. During text generation, it removes the projection of LLM hidden states onto this difference space, thus steering the model’s internal representations toward more human-like patterns.

**Strengths:**

Proposes a novel training-free detection evasion method that constructs a difference space and dynamically edits hidden representations, which is fundamentally different from existing fine-tuning-based methods

The method is highly efficient, requiring only 500 samples to achieve effective evasion

Time cost is significantly reduced (Figure 5 shows orders of magnitude improvement compared to baseline methods)

Includes human evaluation: Maintains good text quality (human evaluation results in Table 2)

**Weaknesses:**

1. The number of human evaluations is limited and could be increased

2. The number of evaluated detectors is limited. Could add:
Fast-DetectGPT, ImBD (designed specifically for detection evasion), Text Fluoroscopy, Binoculars

3. Evaluation scope: Experiments about LLMs to be detected are limited, and only consider two domains : OpenWebText and WritingPrompt

**Questions:**

1. Detection adaptation: Can detectors trained with adversarially augmented data resist R-EAT-type manipulations, like ImBD?

2. Generalization: How robust is the learned difference space when applied to unseen domains or LLM architectures?

---

> ### Author Response · Authors · 2025-11-19
>
> Dear Reviewer,
>
> Thank you for your recognition of our work! Based on your suggestion, we have included the relevant additional experiments and made corresponding revisions to the original manuscript, as detailed below.
>
> ## [Weakness 1] Limited Number of Human Evaluators
> Thank you for your suggestion. We have invited 5 additional human evaluators to score the texts based on fluency, semantic accuracy, and AI generation probability. The results are presented in the table below.
>
> Table 1. Human evalution. The best results are highlighted in bold, while the second-best results are underlined.
>
> ||flu.|Sem.|Prob.|
> |-|-|-|-|
> |DPO-1|$\\underline{3.5438}$|**3.4638**|2.8125|
> |DPO-2|**3.6000**|3.4438|2.7612|
> |HUMPA|2.8075|2.8225|3.1887|
> |ModelEditing|3.4025|3.3013|$\\underline{2.6238}$|
> |R-EAT|3.5436|$\\underline{3.4513}$|**2.5188**|
>
> We have updated the relevant content in **Table 2 (page 8 of the revised file)** based on the results mentioned above.
>
> ## [Weakness 2 and Question 1] Limited Number of detectors
> We have already presented the results for Binoculars in the original manuscript. In response to your suggestion, we have also conducted experiments on three additional detectors: ImBD, Fast-detectGPT, and Text Fluoroscopy. All experimental results have been included in the **Appendix F.2** of the revised pdf file. We present the AUROC and AUPRC results for all baseline methods and our R-EAT on these four detectors, using Llama2-13b across two datasets (OpenWebText and WritingPrompt), as shown in the table below.
>
> Table 2. Detection evasion performance evaluation across four detectors.
>
> |||Fast-detectGPT||ImBD||Text Fluoroscopy||binoculars||Average||
> |-|-|-|-|-|-|-|-|-|-|-|-|
> |||AUROC|AUPRC|AUROC|AUPRC|AUROC|AUPRC|AUROC|AUPRC|AUROC|AUPRC|
> |Openwebtext|Base|0.9808|0.9755|0.9549|0.9580|0.9650|0.9705|0.8998|0.9180|0.9501|0.9555|
> ||HUMPA|0.7249|0.7339|0.4357|0.4355|0.8194|0.8673|0.8789|0.8970|0.7147|0.7334|
> ||Dpo-1|0.6814|0.7775|0.5766|0.6623|0.8753|0.9007|0.9233|0.9276|0.7641|0.8170|
> ||Dpo-2|0.6587|0.7568|0.5619|0.6412|0.8630|0.8909|0.8934|0.9062|0.7443|0.7988|
> ||ModelEditing|0.9943|0.9966|0.9686|0.9717|0.7731|0.8213|0.9287|0.9439|0.9162|0.9334|
> ||R-EAT(Ours)|**0.3379**|**0.4501**|**0.3062**|**0.4103**|**0.6223**|**0.6885**|**0.7504**|**0.8183**|**0.5042**|**0.5918**|
> |WritingPrompt|Base|0.9932|0.9958|0.9943|0.9955|0.9680|0.9780|0.9389|0.9558|0.9736|0.9813|
> ||HUMPA|0.9698|0.9764|0.9453|0.9427|0.8793|0.9036|0.9017|0.9201|0.9240|0.9357|
> ||Dpo-1|0.6593|0.7790|**0.6798**|**0.7580**|0.7527|0.8089|0.8673|0.8781|0.7398|0.8060|
> ||Dpo-2|0.6392|0.7664|0.6905|0.7685|0.7390|0.8024|0.8610|0.8777|0.7324|0.8037|
> ||ModelEditing|0.9834|0.9775|0.9949|0.9961|0.7880|0.8387|0.9102|0.9350|0.9191|0.9368|
> ||R-EAT(Ours)|**0.6294**|**0.7015**|0.7635|**0.7478**|**0.6951**|**0.7741**|**0.7376**|**0.8053**|**0.7064**|**0.7572**|
>
> ## [Weakness 3] Limited Evaluation Scope
> Based on your suggestion, we incorporate two new corpora: **PubMedQA** for biomedical research question answering and **SQuAD** for Wikipedia contexts. For PubMedQA, we only used the questions as prompts to guide the LLM in text generation. All experimental settings remain consistent with those in the original manuscript. The average AUROC and AUPRC results for Llama2-13b on the two new corpora, evaluated **across 11 detectors** (including the 8 detectors from the original manuscript and the 3 new detectors you mentioned), are presented in the table below.
>
> Table 3: Evaluation on PubMedQA and SQuAD across 11 detectors.
>
> |||Base|HUMPA|DPO-1|DPO-2|ModelEditing|R-EAT|
> |-|-|-|-|-|-|-|-|
> |PubMedQA|AUROC|0.8367|0.7236|0.8249|0.8167|0.7594|**0.5948**|
> ||AUPRC|0.8720|0.7692|0.8609|0.8550|0.8012|**0.6684**|
> |SQuAD|AUROC|0.8991|0.8436|0.8851|0.8723|0.8247|**0.5899**|
> ||AUPRC|0.9113|0.8541|0.9051|0.8979|0.8557|**0.6558**|
>
> The results in the table show that our R-EAT achieves the best average evasion performance across 11 detectors on the two new datasets. Detailed results can be found in **Appendix F.1** of the revised manuscript.

---

> ### Author Response · Authors · 2025-11-19
>
> ## [Question 2] Generalization
> In Appendix E of the manuscripts (**Appendix F.3 in revised manuscript**), we present the results of constructing the difference space on Llama2/OpenWebText and applying it to guide text generation on the WritingPrompts dataset (denoted as 'R-EAT$\_{ood}$'). The results show that R-EAT$\_{ood}$ even outperforms the baseline method trained directly on WritingPrompts, with performance only slightly lower than constructing the space using WritingPrompts itself.
> we provide the average AUROC across 8 detectors, as shown in the Table 4 below.
>
> Table 4. Average AUROC of the Generalization Ability of R-EAT. The best results are highlighted in bold, while the second-best results are underlined.
>
> | |Base|HUMPA|DPO-1|DPO-2|ModelEditing|R-EAT|R-EAT$\_{ood}$|
> |---|---|---|---|---|---|---|---|
> |Average AUROC|0.8675|0.8798|0.7045|0.7019|0.8920|**0.6879**|$\\underline{0.7004}$|
>
> We hope our responses have addressed your concerns adequately. If you have any further questions or would like to discuss any aspects of our work, we would be happy to continue the discussion.

---

### Official Review · Reviewer_qYdx · 2025-10-29

**Soundness:** 3
**Presentation:** 3
**Contribution:** 3
**Rating:** 6
**Confidence:** 4

**Summary:**

The paper proposes training-free approach to make AI-generated text less detectable by AI-text detectors. R-EAT constructs a difference space between AI-generated and human-written texts, identifies discriminative features via singular value decomposition (SVD), and removes AI-like components from hidden states to generate more human-like text. This method avoids the computational cost of fine-tuning, improves sample and time efficiency, and reduces the detectability of AI text across multiple detectors while preserving text quality.

**Strengths:**

1. The proposed idea is innovative.

2. The paper is clearly structured and well written.

3. The theoretical analysis is sound and well justified.

**Weaknesses:**

1. R-EAT relies on a set of human-written and AI-generated texts to derive the difference space. Gathering representative samples of both HWT and AIGT are needed to compute the key representation differences.

2. The attack must be implemented inside the model’s generation process, requiring internal access to the LLM’s hidden states at a particular layer. However, the frontier models are API models. How effective or useful this method will be is still in question.

3. Could the method evade fast-detectgpt[R1]?

[R1]. Bao, G., Zhao, Y., Teng, Z., Yang, L., & Zhang, Y. (2023). Fast-detectgpt: Efficient zero-shot detection of machine-generated text via conditional probability curvature. arXiv preprint arXiv:2310.05130.

**Questions:**

See weaknesses

---

> ### Author Response · Authors · 2025-11-19
>
> Dear Reviewer,
>
> Thank you for your recognition of our work! We have provided detailed responses to each of your concerns below.
> ## [Weakness 1] Dependence on Representative Samples of HWT and AIGT.
> Thank you for your thorough review! In our experiments, we used the first 500 samples from the training set to construct the difference space. To quantitatively analyze the impact of different samples, we used 3 different random seeds to select the samples and reevaluated across all detectors. The results are shown in the table below.
>
> Table 1. Evasion Detection Performance Across Different Random Seeds (Mean/Standard)
>
> ||Average AUROC (mean/std)|Average AUPRC (mean/std)|
> |-|-|-|
> |Llama2-13b/openwebtext|0.6477 / 0.0102|0.7140 / 0.0114|
>
> The results in the table above demonstrate that R-EAT is capable of extracting effective spaces from a small number of samples and exhibits robustness to sample noise.
>
> ## [Weakness 2] Applicability under the API Model
> Exploring evasion detection methods in API model scenarios is both important and meaningful. We acknowledge that R-EAT requires a fully accessible LLM to construct the difference space and guide the modification of hidden representations. In API-model scenarios, we can only rely on carefully designed prompts to steer the LLM toward generating more human-like text. In this context, **R-EAT offers an interpretable framework for guiding prompt design.**
>
> As analyzed in Figure 1 on page 2 of the revised pdf file, all evasion-based methods—such as LLM fine-tuning, model editing, representation editing (our R-EAT), and prompt design in API model settings—essentially aim to alter the hidden representations. Building on this observation, **we suggest that the effectiveness of different evasion prompts can be systematically evaluated by examining how their induced hidden representations project onto the difference space**, for example using a proxy fully accessible LLM for analysis. This insight enables researchers to optimize prompt designs and identify the most effective strategies. In the future, we will continue to explore this issue in greater depth.
>
> ## [Weakness 3] Additional experiments on Fast-detectGPT
> Based on your suggestion, we have included additional results showing the performance of R-EAT and all baseline methods on Fast-detectGPT using Llama2-13b, as presented in the table below.
>
> Table 2. Performance evalution on Fast-detectGPT
>
> ||Openwebtext||||||
> |-|-|-|-|-|-|-|
> ||Base|HUMPA|Dpo-1|Dpo-2|ModelEditing|R-EAT|
> |AUROC|0.9808|0.7249|0.6814|0.6587|0.9943|**0.3379**|
> |AUPRC|0.9755|0.7339|0.7775|0.7568|0.9966|**0.4501**|
> ||WritingPrompt||||||
> ||BASE|HUMPA|Dpo-1|Dpo-2|ModelEditing|R-EAT|
> |AUROC|0.9932|0.9698|0.6593|0.6392|0.9834|**0.6294**|
> |AUPRC|0.9958|0.9764|0.7790|0.7664|0.9775|**0.7015**|
>
> The results indicate that our R-EAT outperforms all baseline methods in effectively evading Fast-detectGPT. Experimental results are provided in **Appendix F.2** of the revised manuscript.
>
> We hope our responses have addressed your concerns adequately. If you have any further questions or would like to discuss any aspects of our work, we would be happy to continue the discussion.

---

> ### Comment · Reviewer_qYdx · 2025-11-27
>
> Thanks the author for answering my questions.

---

### Official Review · Reviewer_zLaZ · 2025-10-30

**Soundness:** 2
**Presentation:** 3
**Contribution:** 2
**Rating:** 2
**Confidence:** 3

**Summary:**

This paper aims to evaluate the robustness of AI-generated text (AIGT) detectors and proposes a novel method for detection evasion. The authors argue that existing evasion methods have limitations: fine-tuning-based approaches (like DPO) demand significant computational and data resources , while model editing methods (which directly modify weights) suffer from limited performance due to the residual connections in the Transformer architecture.

To address these issues, the paper introduces R-EAT (Representation Editing Attack), a "training-free" evasion method. The core mechanism of R-EAT is:

* Constructing a Difference Space: First, it collects hidden layer representations of AIGT and human-written text (HWT) at specific layers of the LLM to construct a "difference space" between them.
* Dynamic Editing: During the text generation process, R-EAT dynamically modifies the LLM's hidden states by removing their projections onto this difference space . The authors claim this guides the model to produce more "human-like" text.

The main contributions of this paper include:
* Proposing R-EAT, a training-free attack method based on representation editing.
* Providing theoretical analysis arguing that R-EAT, by directly editing hidden states, avoids the inherent limitations of model editing methods while offering superior sample and time efficiency compared to DPO-based fine-tuning.
* Presenting experimental results showing that R-EAT effectively reduces the average detection accuracy (AUROC) of eight AIGT detectors on texts generated by two LLMs (Llama2-13b and Qwen3-14b). The paper also reports advantages in time efficiency, sample requirements (using 500 samples in experiments), and preservation of text quality.

**Strengths:**

This paper presents a novel and efficient method for evading AI-generated text (AIGT) detection, supported by theoretical analysis and a comprehensive experimental setup.

* Originality: The primary strength of this paper is its originality. The proposed method, R-EAT (Representation Editing Attack), introduces a new paradigm for AIGT detection evasion. While prior work has focused on post-processing , resource-intensive fine-tuning (e.g., DPO) , or modifying model weights , this is the first work, to my knowledge, to propose directly editing the hidden representations during the generation process to achieve evasion. This adaptation of representation editing principles to the AIGT evasion task is a creative and novel contribution.

* Clarity: The paper is well-written and clearly organized. The core concept is introduced intuitively with helpful diagrams (especially Figure 1, which clearly contrasts the R-EAT paradigm with previous methods). The methodology in Section 3.2 is broken down into two distinct, easy-to-understand steps: "Difference Space Construction" and "Representation Editing". The mathematical notation is clear and consistent.

**Weaknesses:**

While this paper proposes a novel method, it suffers from several critical flaws that undermine the reliability of its conclusions and the robustness of its contributions.

* Serious Deficiencies in Literature Review: The "Related Work" section (Section 2) fails to properly situate R-EAT within the critical discourse on AIGT detection evasion. The authors have omitted foundational work that discusses the fundamental limits of detection and adversarial robustness. For instance, (Sadasivan et al., 2023, "On the possibilities of AI-generated text detection") and (Hu et al., 2023, "RADAR: Robust AI-Text Detection via Adversarial Learning") are core literature in this field.

  * [1] makes the fundamental argument that as LLMs themselves improve, AIGT detection may become theoretically infeasible. The paper, while claiming successful evasion, does not engage with this core challenge.
  * [2] specifically investigates hardening detectors against attacks via adversarial learning (e.g., RADAR). This paper proposes a new attack but completely fails to evaluate it against known robust defenses like RADAR.

   This omission is severe. It is not just a missing citation; it indicates the authors may not have benchmarked their work against the most important theoretical and defensive research in the field, leading to an overstatement of the contribution's context and significance.


**References**

[1] Souradip Chakraborty, Amrit Singh Bedi, Sicheng Zhu, Bang An, Dinesh Manocha, and Furong
Huang. On the possibilities of ai-generated text detection.

[2] Xiaomeng Hu, Pin-Yu Chen, Tsung-Yi Ho. RADAR: Robust AI-Text Detection via Adversarial Learning.

**Questions:**

Given that R-EAT is proposed as an attack method, its evaluation against standard detectors is insufficient for a complete robustness analysis. Can the authors provide additional experiments comparing R-EAT's effectiveness against detectors specifically designed to be robust to adversarial attacks, such as RADAR ?

**Details Of Ethics Concerns:**

no concerns.

---

> ### Author Response · Authors · 2025-11-19
>
> Dear Reviewer,
>
> We sincerely appreciate your recognition of the novelty and paper writing of our work. Regarding the shortcomings you pointed out in our literature review, we have revised Section 2.1 and related appendix in the updated version accordingly. In addition, we have carefully analyzed the two referenced works you mentioned and provide our detailed responses below.
>
> ## [1] Chakraborty, Souradip, et al. "Position: On the possibilities of AI-generated text detection." Forty-first International Conference on Machine Learning. 2024.
>
> **We fully acknowledge the contributions of the paper** you referenced to the advancement of the field. We agree with the perspectives presented and would be happy to engage in further discussion on them.
>
> First, one point worth noting is the **apparent contradiction** between the viewpoint in the work of Sadasivan et al [3] mentioned in your weakness section and the arguments in the referenced paper. **In Sadasivan et al.’s paper**, **Theorem 1** establishes an upper bound on the area under the ROC curve (AUROC) of any detector D. This theorem asserts that if the model is sufficiently good, the statistical difference between the AI and human outputs TV(m,h) will approach zero, making it theoretically impossible for any detector to perform better than random guessing.
> However, **this viewpoint is challenged** in the subsequent paper referenced in your review, "**On the Possibilities of AI-generated Text Detection**". In Section 3 of this paper, **both Theorem 1 and Theorem 2** demonstrate that as long as the two distributions are not exactly the same, which is rarely the same, the **detection will always be possible** by collecting more samples as established next.
>
> Second, we argue that **it is practically impossible for the text distribution of a language model to be completely identical to that of human writing.** A language model $P(m)$ can only learn from a finite training dataset $D\_{train}$, and thus can never fully capture all the latent variations in the true data-generating process, i.e., human writing $P(h)$. Therefore, $P(m)$ can only serve as an approximation of $P(h)$. Current LLMs are trained on large amounts of human-generated text, yet AIGT detection remains a hot topic of interest within the research community.
>
> Finally, we also acknowledge that with the development of LLMs, the distributional gap between AIGT and HWT will continue to shrink, making the detection of AIGT increasingly difficult. However, each step of LLM advancement requires substantial computational and data resources. Our proposed R-EAT method offers an efficient (both in terms of time and sample efficiency) attack strategy, providing a practical means to evaluate and stress-test detection methods under increasingly realistic conditions. This, in turn, supports the development of more robust and generalizable detection approaches.
>
> *[3] Sadasivan, Vinu Sankar, et al. "Can AI-generated text be reliably detected? stress testing AI text detectors under various attacks." Transactions on Machine Learning Research (2025).*

---

> ### Author Response · Authors · 2025-11-19
>
> ## [2] Hu, Xiaomeng, Pin-Yu Chen, and Tsung-Yi Ho. "Radar: Robust ai-text detection via adversarial learning." Advances in neural information processing systems 36 (2023): 15077-15095.
> Following previous work [4], we have considered several representative detection methods from recent years, most of which are state-of-the-art zero-shot detection approaches, including Binoculars (ICML'24), Lastde and Lastde++ (ICLR'25), among others. Additionally, **we fully agree with your point that studying the results against existing adversarial detectors can significantly extend the relevance of our method.** Therefore, we conduct experiments on two defensive techniques, Radar (based on your suggestion) and ImBD (the latter specifically designed for detection evasion, as suggested by Reviewer #26KA), and provide the results on Llama2-13b/OpenWebText as follows.
>
> Table 1. Evaluation of different evasion detection methods on defensive detectors
>
> ||Radar||ImBD||Average||
> |--|--|-|-|-|-|-|
> ||AUROC|AUPRC|AUROC|AUPRC|AUROC|AUPRC|
> |Base|0.9886|0.9892|0.9549|0.9580|0.9718|0.9736|
> |HUMPA|0.9993|0.9993|0.4357|0.4355|0.7175|0.7174|
> |DPO-1|**0.9080**|**0.9150**|0.5766|0.6623|0.7423|0.7887|
> |DPO-2|0.9109|0.9183|0.5619|0.6412|0.7364|0.7798|
> |ModelEditing|0.9809|0.9834|0.9686|0.9717|0.9748|0.9776|
> |R-EAT|0.9408|0.9530|**0.3062**|**0.4103**|**0.6235**|**0.6817**|
>
> It can be observed that R-EAT leads to the greatest decrease in the average detection AUROC for the two robust detectors, dropping from 97.18% to 62.35%
> In terms of Radar, although R-EAT performs slightly worse than DPO-1 and DPO-2, it outperforms HUMPA and model-editing methods. Considering that DPO1 and DPO2 are trained on much larger datasets compared to R-EAT, we performed a comparison using the same dataset size (500 samples). As shown in Table 2, when using the same amount of data, R-EAT outperforms both DPO1 and DPO2 by a significant margin.
>
> Table 2. Evaluation of performance on Radar using different methods with 500 samples.
>
> ||Radar||
> |-|-|-|
> ||AUROC|AUPRC|
> |DPO-1|0.9607|0.9640|
> |DPO-2|0.9676|0.9656|
> |R-EAT|**0.9408**|**0.9530**|
>
> An interesting observation is that, while statistical detection methods have gained significant attention in recent years due to their training-free nature, they still underperform in the face of adversarial attacks like R-EAT when compared to supervised methods such as Radar. This highlights a key limitation of training-free approaches, underscoring the need for more robust detection strategies that can better handle sophisticated attacks. Additionally, we also evaluate the performance against Radar on two new datasets. All detailed experimental results are provided in Appendix F.1 and F.2.
> This finding aligns with our goal, as emphasized in the Ethical Statement section: "Our goal is to understand model behaviors and identify limitations in current detection techniques, ultimately guiding the development of more robust detectors."
>
> *[4] Wang, Tianchun, et al. "Humanizing the Machine: Proxy Attacks to Mislead LLM Detectors." The Thirteenth International Conference on Learning Representations 2025.*
>
> We hope our responses have addressed your concerns adequately. If you have any further questions or would like to discuss any aspects of our work, we would be happy to continue the discussion.

---

> > ### Comment · Reviewer_zLaZ · 2025-11-27
> >
> > I thank the authors for their comprehensive response and for conducting the suggested experiments against adversarial detectors (RADAR and ImBD). The inclusion of these results significantly improves the robustness analysis and properly situates R-EAT within the current state-of-the-art defenses.
> >
> > I find the additional analysis on data efficiency (Table 2) particularly persuasive. It effectively highlights the specific advantage of R-EAT, achieving competitive evasion performance with minimal data (500 samples), which distinguishes it from resource-intensive methods like DPO. While I note that RADAR maintains a high detection rate against R-EAT (AUROC ~0.94), the trade-off offered by your method in terms of efficiency and training-free operation is a valuable contribution.
> >
> > Since my primary concerns regarding the missing baselines and literature gap have been adequately addressed, I have raised my score to reflect the improved completeness of the paper.

---

> > > ### Author Response · Authors · 2025-11-28
> > >
> > > We sincerely thank the reviewer for the invaluable feedback and for recognizing the improvements introduced by our additional experiments. We also appreciate the reviewer’s insightful suggestion regarding the gap in the related literature. The inclusion of these relevant works and baseline comparisons has made our study more comprehensive and reliable.

---

### Official Review · Reviewer_4Y1b · 2025-11-10

**Soundness:** 3
**Presentation:** 3
**Contribution:** 3
**Rating:** 6
**Confidence:** 2

**Summary:**

The paper introduces R-EAT, a training-free approach for evading AI-generated text detectors. Distinct from previous finetuning- or model-editing-based evasion methods, R-EAT directly modifies hidden representations of large language models by removing projections onto discriminative directions that separate human-written and AI-generated text, constructed via SVD.

**Strengths:**

R-EAT is a training-free approach that avoids the substantial computational and data costs seen in finetuning- and proxy-based evasion methods.

The theoretical section, especially Equations 11 and 12 in Section 3.3, details why direct representation editing is fundamentally more potent than model editing due to the architectural impact of residual connections.

**Weaknesses:**

1. **Clarity of Mathematical Specification**: While the main formulas are present, Sec 3.2 contains ambiguous variable indexing and notational inconsistencies. For example, the definition of $\bar{\boldsymbol{h}}_j$ and the mean computation for the difference space would benefit from clean, explicit sample definitions especially wrt. set sizes $N^+$ vs $N$. The explanation around the centering step in the difference space construction is terse and glosses over rationale for removing mean direction, which could confuse readers less familiar with representation alignment methods.

2. **Hyperparameter Sensitivity Not Fully Explored**: Although Fig 6–9 show R-EAT's behavior under changing $\tau$, $\alpha$, and layers, the explanations lack depth on why particular values perform best, and there is little practical guidance for practitioners. For example, Fig 7 shows a non-monotonic trend in AUROC versus $\alpha$, but the text does not explain what types of distortion or artifacts arise at higher editing strengths.

3.**Limited Discussion of Generalization and Transfer:** While R-EAT performs well on the presented Llama2-13b and Qwen3-14b models (Section 4.5), the method heavily relies on representation differences between AIGT and HWT at specific layers in these models. There is little discussion or empirical evaluation of how learned difference spaces transfer to unseen models or domains (other than an Appendix pointer); thus, practical applicability in real-world black-box or rapidly-evolving deployment settings is underexplored.

**Questions:**

How robust is the learned difference space for R-EAT when used on models or text domains not represented in the preference dataset? For instance, can the same $\boldsymbol{B}_j$ constructed on Llama2-13b/OpenWebText be used for Qwen3-14b or other text sources with minimal loss of efficacy?

Given that R-EAT is proposed as a research contribution, what steps are planned (if any) to control misuse, especially if code/models are released? Are there watermarking, detection countermeasures, or disclosure policies planned?

---

> ### Author Response · Authors · 2025-11-19
>
> Dear Reviewer，
>
> Thank you very much for your positive feedback on our work! We have carefully addressed each of your comments and made the corresponding revisions in the revised PDF file, where the revised sentences are highlighted in blue. We hope our responses have resolved your concerns. If you have any further questions or suggestions, we would be happy to discuss them with you.
>
> ## [weakness 1] Clarity of Mathematical Specification.
> - We clarified the definition of **$ \bar{h}\_j $**, specifying it as the mean vector of  **$h\_{i,j}^+$** $(i=1,...,N)$. **(line 197 of the revised PDF file)**
> - We removed the description of $N\^⁺$ and replaced it with $N$. **(line 197 of the revised PDF file)**
> - We provided a more detailed explanation of the computation of the centered difference matrix during the construction of the difference space, and added this clarification to the **line 193 and Appendix B of the revised PDF file**. We find that the mean directions of **$H\_j\^+$** and **$H\_j\^-$** are aligned with their respective un-centered top right singular vectors, and that these mean directions are also aligned with each other. Therefore, we assume that the **$H\_j\^+$** and **$H\_j\^-$** share the same mean direction. To focus on the directions that are orthogonal to the overall human representation, we remove the component along the mean direction of **$H\_j\^+$**.
>
> We sincerely thank the reviewer again for the insightful comments. We believe these revisions have made our paper more accurate and have improved its overall quality.
> ## [weakness 2] Hyperparameter Sensitivity Not Fully Explored.
> Based on your suggestion, we conduct further analysis for hyperparameter sensitivity.
> - **Impact of the cumulative variance threshold $\\tau$ (line 467 of the revised PDF file).**
>
> An increase in $\\tau$ leads to a greater number of singular vectors used to construct the difference space, enabling the space to capture the distinctions between HWT and AIGT more accurately, which in turn improves evasion performance.
> - **Impact of editing strength $\\alpha$ (line 472 of the revised PDF file).**
>
> This parameter $\\alpha$ controls the proportion of the target directional component removed from original hidden representation $h\_{new,j}$. When $\\alpha=0$, the original state is fully preserved, while when $\alpha=1$, the component along the direction of the difference space is entirely removed. Initially, increasing $\\alpha$ effectively improves the evasion performance of the generated text. However, when the editing strength exceeds a certain threshold (e.g., when $\\alpha=0.7$ as shown in the Figure 7 on page 9 of the revised PDF file.), the semantic integrity of the generated text is affected, leading to a detection evasion performance decline. We evaluate the $|\\Delta PPL|$ of the generated texts at different editing strengths, which supports our hypothesis, as shown in table below.
>
> Table 1. Impact of editing strength $\\alpha$
> | $\\alpha$|$\|\\Delta PPL\|$|
> |---:|---:|
> |0.1|30.104|
> |0.3|26.013|
> |0.5|4.5814|
> |0.7|1.7057|
> |0.9|7.9373|
> |1.0|3.5539|
>
> We have included this analysis in **Appendix F.6 of the revised pdf file.**
> - **Impact of the number of layers edited (line 489 of the revised pdf file).**
>
> Through a more detailed analysis of the impact of the number of edited layers on evasion performance, we observe the following trends:
> **For single-layer editing**, as the layer depth increases, the evasion performance improves. This is because when the edited hidden vectors are passed to the subsequent layers, they are still influenced by the model features encoded in the deeper layers, which ultimately degrades the evasion performance. The effect is more pronounced when the edited layer is closer to the beginning of the model. Interestingly, when only the last layer’s hidden vectors are edited, the performance actually worsens. We believe this occurs because the last layer mainly maps hidden representations to output probability distributions, and modifying it alone only slightly adjusts the output without significantly altering the model’s generation strategy, which limits the effectiveness of the evasion.
>
> Additionally, **for multi-layer editing**, a clear trend emerges: the more layers that are edited, the better the evasion performance, which aligns with the cumulative effect observed in Section 3.2. However, we also observe that editing both layers 38 and 39 results in worse performance compared to editing only a single layer. We attribute this phenomenon to the large edits introduced at layer 38, which cause higher-order terms in the nonlinear transformations to become significant. This aligns with the remark made in Appendix C.1.

---

> ### Author Response · Authors · 2025-11-19
>
> - **Impact of the number of samples (Line 478 of the revised pdf file).**
>
> Based on the observation in Figure 8 on page 9 of the revised pdf file, the best evasion performance is achieved when constructing the difference space with 100 samples. However, as the sample size increases, the evasion performance starts to degrade. To analyze this phenomenon, we conduct a detailed investigation by extracting **the number of singular vectors** required for constructing the difference space at different sample sizes, as well as calculating **the cosine of the principal angle** between the difference space constructed with multiple samples and the one built with 100 samples.
>
> As shown in Table 2 below, we find that **as the sample size increases, the number of singular vectors satisfying  $\\tau>90%$ also increases.** This indicates a significant rise in the dimensionality and complexity of the difference space. Additionally, we observe that **the cosine of the principal angle between the difference space constructed with 200 samples and the one built with 100 samples is the lowest.** This suggests that the difference space at this point introduces a significant amount of suboptimal features and sample-specific noise, leading to a substantial decline in evasion performance. Moreover, with the further increasing sample size, although the direction of the difference space begins to revert toward the primary direction, the overall dimensionality of the difference space continues to increase, causing the proportion and absolute amount of noisy features to rise. Consequently, the evasion performance starts to recover, but it still remains below the optimal level achieved with 100 samples.
>
> Table 2. Impact of the number of samples
> |Sample number|Number of singular vectors|Max principal angle cosine|
> |---:|---:|---:|
> |100|10|-|
> |200|125|0.1884|
> |300|180|0.2087|
> |400|236|0.2465|
> |500|285|0.2739|
> ## [weakness 3 and Question 1] Limited Discussion of Generalization and Transfer.
> - **Generalization to Unseen Models or Corpora (line 401)**
>
> Based on your suggestions, we have expanded the discussion of R-EAT's generalization in Section 4.
> We evaluate the generalization of the difference space across different corpora. Specifically, we construct the difference space on Llama2/OpenWebText and apply it to guide text generation on the WritingPrompts dataset (denoted as R-EAT$\_{ood}$).
> The results show that R-EAT$\_{ood}$ even outperforms the baseline method trained directly on WritingPrompts, with performance only slightly lower than constructing the space using WritingPrompts itself. These results show that R-EAT has strong generalization ability.
>
> Table 3. Average AUROC of the Generalization Ability of R-EAT. The best results are highlighted in bold, while the second-best results are underlined.
>
> | |Base|HUMPA|DPO-1|DPO-2|ModelEditing|R-EAT|R-EAT$\_{ood}$|
> |---|---|---|---|---|---|---|---|
> |Average AUROC|0.8675|0.8798|0.7045|0.7019|0.8920|**0.6879**|$\\underline{0.7004}$|
>
> - **Applicability under the API Model**
>
> Exploring evasion detection methods in API model scenarios is both important and meaningful. We acknowledge that R-EAT requires a fully accessible LLM to construct the difference space and guide the modification of hidden representations. In API-model scenarios, we can only rely on carefully designed prompts to steer the LLM toward generating more human-like text. In this context, **R-EAT offers an interpretable framework for guiding prompt design.**
>
> As analyzed in Figure 1 on page 2 of the revised pdf file, all evasion-based methods—such as LLM fine-tuning, model editing, representation editing (our R-EAT), and prompt design in API model settings—essentially aim to alter the hidden representations. Building on this observation, **we suggest that the effectiveness of different evasion prompts can be systematically evaluated by examining how their induced hidden representations project onto the difference space**, for example using a proxy fully accessible LLM for analysis. This insight enables researchers to optimize prompt designs and identify the most effective strategies. In the future, we will continue to explore this issue in greater depth.

---

> ### Author Response · Authors · 2025-11-19
>
> ## [Question 2] Potential approaches to prevent misuse
> **This is indeed an important issue.** As mentioned in Appendix G, we have discussed several possible defense strategies, such as **adversarial training** and **ensemble detection** that combines multiple detectors. In addition, **we consider watermarking to be a promising approach for preventing misuse.** Existing watermarking methods typically operate at the **logit level** by introducing biases to token probabilities, thereby enforcing specific generation patterns. For example, in the red–green watermarking approach, tokens are divided into “red” and “green” lists. Although R-EAT alters the original probability distribution during generation, it is still possible to preserve watermark detectability by adjusting the token selection process to further favor tokens from the green list. As a result, the majority of generated tokens fall within the green list, allowing the watermark to remain detectable.

---

### Note · Program_Chairs · 2026-01-17
**Submission Desk Rejected by Program Chairs**

The following references in this submission do not refer to real documents and/or have major errors in bibliographic information:

 Nicolò Pedrotti, Alessio Miaschi, Giovanni Puccetti, Felice Dell'Orletta, and Giulia Venturi. The unbearable weight of machine-generated text: A stress test for aigt detectors. In Findings of the Association for Computational Linguistics: ACL 2024, pp. 833-846. Association for Computational Linguistics, 2024. URL https://aclanthology.org/2024.findings-acl.54.

Haoran Li, Jun-Jie He, Wen-Fan Shen, Shi-Biao Qiu, Wei-Shi Gao, Ka-Man Huang, Yuan-Yuan Cao, and Yue-Ga Huang. Lastde: A local-aggregated and sampling-based transformer decoder for text detection. In Proceedings of the IEEE/CVF Conference on Computer Vision and Pattern Recognition, pp. 28003-28013, 2024.